# A lncRNA Dleu2-encoded peptide relieves autoimmunity by facilitating Smad3-mediated Treg induction

Sibei Tang[1,3], Junxun Zhang[2,3], Fangzhou Lou[1], Hong Zhou [iD][1], Xiaojie Cai [iD][1], Zhikai Wang[2], Libo Sun [iD][1], Yang Sun[1], Xiangxiao Li[1], Li Fan[1], Yan Li[1], Xinping Jin[2], Siyu Deng[1], Qianqian Yin[2], Jing Bai[1], Hong Wang[2] & Honglin Wang [iD][1✉]

## Abstract

Micropeptides encoded by short open reading frames (sORFs) within long noncoding RNAs (lncRNAs) are beginning to be discovered and characterized as regulators of biological and pathological processes. Here, we find that lncRNA Dleu2 encodes a 17-amino-acid micropeptide, which we name Dleu2-17aa, that is abundantly expressed in T cells. Dleu2-17aa promotes inducible regulatory T (iTreg) cell generation by interacting with SMAD Family Member 3 (Smad3) and enhancing its binding to the *Foxp3* conserved non-coding DNA sequence 1 (CNS1) region. Importantly, the genetic deletion of Dleu2-17aa in mice by start codon mutation impairs iTreg generation and worsens experimental autoimmune encephalomyelitis (EAE). Conversely, the exogenous supplementation of Dleu2-17aa relieves EAE. Our findings demonstrate an indispensable role of Dleu2-17aa in maintaining immune homeostasis and suggest therapeutic applications for this peptide in treating autoimmune diseases.

**Keywords** LncRNA Encoded Micropeptide; Autoimmunity; Treg Cell; Dleu2; Smad3
**Subject Categories** Immunology; RNA Biology; Signal Transduction

## Introduction

Autoimmune diseases are characterized by overreactive immune responses towards self-antigens, leading to a breakdown in self-tolerance. A critical player in maintaining immune tolerance is the CD4+FOXP3+ regulatory T (Treg) cell that suppresses effector T cell responses (Sakaguchi et al, 2010; Itoh et al, 1999; Ziegler, 2006). Treg cell deficiency or dysfunction has been observed in various autoimmune diseases, such as multiple sclerosis, diabetes, and rheumatoid arthritis (Viglietta et al, 2004; Lindley et al, 2005; Ehrenstein et al, 2004). Therefore, expanding the Treg cell population or enhancing their function represents a promising therapeutic strategy for managing autoimmune diseases.

Forkhead box P3 (FOXP3) is the central determinant of Treg development and function, which is expressed constitutively in thymus-derived natural Treg (nTreg) cells and inducibly in peripheral Treg (pTreg or iTreg) cells (Viglietta et al, 2004; Lindley et al, 2005; Ehrenstein et al, 2004). A crucial pathway for *Foxp3* gene expression and function is the transforming growth factor β (TGF-β)-mediated SMAD Family Member (Smad) signaling. Upon TGF-β binding to its receptors, SMAD Family Member 2 (Smad2) and SMAD Family Member 3 (Smad3) are activated and form a complex with SMAD Family Member 4 (Smad4); this complex translocates to the nucleus and binds to the *Foxp3* conserved noncoding DNA sequence (CNS1) region, initiating *Foxp3* transcription (Schlenner et al, 2012; Lagna et al, 1996; Tone et al, 2008). Several factors, including retinoic acid, have been shown to enhance the TGF-β/Smad3-dependent induction of Treg cells (Xiao et al, 2008), which highlights the importance of the TGF-β/Smad3 pathway in Treg cell-mediated immune homeostasis and the therapeutic potential of manipulating this pathway to treat autoimmune diseases.

Long noncoding RNAs (lncRNAs) are RNA transcripts longer than 200 nucleotides that lack long protein-coding open reading frames (ORFs) (Nagano and Fraser, 2011). Recent advances in ribosome profiling studies have shown that some lncRNA transcripts contain short ORFs (sORFs) that code for functional micropeptides, which participate in various cellular activities (Magny et al, 2013; Anderson et al, 2015; Bi et al, 2017; Zhu et al, 2020; Matsumoto et al, 2017; Huang et al, 2021; Chugunova et al, 2019, 00116; Stein et al, 2018; Wang et al, 2020; Niu et al, 2020; Ge et al, 2021; Guo et al, 2020; Zhang et al, 2022; Bhatta et al, 2020; Wu et al, 2022). We and others have revealed the immunoregulatory roles of several micropeptides. We reported that human *MIR155HG* encodes miPEP155 that suppresses antigen presentation in dendritic cells during autoimmunity

[1]Precision Research Center for Refractory Diseases, Shanghai General Hospital, Shanghai Jiao Tong University School of Medicine, Shanghai 201610, China. [2]Shanghai Institute of Immunology, Shanghai Jiao Tong University School of Medicine, Shanghai 200025, China. [3]These authors contributed equally as first authors: Sibei Tang, Junxun Zhang.
✉E-mail: honglin.wang@sjtu.edu.cn

(Niu et al, 2020); mouse *mir31hg* encodes miPEP31 that acts as a transcriptional repressor, inhibiting the expression of the pro-inflammatory miRNA-31 (Zhou et al, 2022). Recent studies showed that *1810058I24Rik*-encoded micropeptide Mm47 participates in the activation of the Nlrp3 inflammasome (Bhatta et al, 2020, 47), and *hemotin* encodes a micropeptide that serves as a phagocytic regulator in vertebrates (Pueyo et al, 2016). These micropeptides are promising drug candidates owing to good cell-penetrating properties, strong activities, and minimal antigenicity or side effects (Vasconcelos et al, 2013; Wang et al, 2022; Muttenthaler et al, 2021). We thus intended to discover more endogenously-existing peptides with immunomodulatory functions.

In this study, we identified an endogenous micropeptide, Dleu2-17aa, which was encoded by the lncRNA Deleted in Lymphocytic Leukemia 2 (*Dleu2*). We show that Dleu2-17aa directly interacted with Smad3, which promoted its binding to the CNS1 region of the *Foxp3* gene and led to an increase of Foxp3 expression and Treg cell generation. We generated start codon-mutated mice with Dleu2-17aa genetically ablated while the expression of its host gene unaffected and found that these mice were defective in generating iTreg cells and were thus more susceptible to experimental autoimmune encephalomyelitis (EAE). We further showed that exogenously delivered Dleu2-17aa significantly alleviated EAE, which demonstrated its therapeutic potential for treating auto-immune diseases.

# Results

## Dleu2 encodes a novel endogenous micropeptide, Dleu2-17aa

In a bioinformatic screening for possible immune-functional lncRNA-encoding micropeptides using ORFfinder, we discovered that the *Dleu2* gene in mice may harbor a sORF. Analysis of the Dleu2 RNA transcripts revealed a predicted sORF encoding a 17-amino acid, located in the first exon of the Dleu2 RNA transcripts on chromosome 14 (Fig. 1A), a similar 25 amino acid sORF was found in human DLEU2 transcripts, which we referred as Dleu2-17aa and DLEU2-25aa respectively (Fig. 1B). These two micro-peptides reside within the 5′-terminal region of their respective RNA transcripts, where proved to be the only and the most conserved region of these two RNA among mouse (NR_028264.1) and human (NR_152566.1) transcripts (Appendix Fig. S1).

To further investigate the protein synthesis potential of Dleu2-17aa ORF, we analyzed the aggregated ribosome profiling data (Han et al, 2019; Jackson et al, 2018; Xu et al, 2019) with the Integrative Genomics Viewer (IGV). We observed a strong ribosome footprint signal that overlapped with the Dleu2-17aa ORF, indicating its translational activity (Fig. 1C). We then fused the wild-type (WT) or mutant (ATG to ATT) Dleu2-17aa ORF with EGFP to determine the translational activity of the Dleu2-17aa ORF (Fig. 1D). Fluorescence imaging (Fig. 1E) and western blot analysis (Fig. 1F) showed only the WT ORF-EGFP construct produced detectable EGFP expression. This confirms the actual translation of the Dleu2-17aa ORF.

To elucidate the endogenous expression of Dleu2-17aa, we generated a polyclonal antibody against it with high sensitivity and specificity (Appendix Fig. S2). We performed real-time PCR (Fig. 1G) and

immunoblotting (Fig. 1H) analysis and found that Dleu2-17aa was extensively expressed in immune organs, including the lymph node (LN), spleen (SP), and thymus (Thy). These results indicated that mouse *Dleu2*, annotated as lncRNA, encodes a 17-amino acid endogenous micropeptide, which may participate in immune regulation.

## Dleu2-17aa promotes Treg cell differentiation in vitro

To investigate the properties and immunological functions of Dleu2-17aa, we first assessed the expression pattern of Dleu2-17aa in various immune cells, including B cells, T cells, dendritic cells (DC), and macrophages (Mφ). Our analysis revealed that Dleu2-17aa expression was highest in T cells, suggesting its potential involvement in T cell biology (Fig. 2A,B).

To elucidate the function of Dleu2-17aa in T cells, we conjugated it with a 6-FAM fluorophore and examined its penetrating ability. Our data showed its capacity to penetrate into different types of T cells, including EL4 cells, Jurkat cells (Fig. EV1A), and mouse CD4$^+$ T cells (Fig. EV1B,C). We observed dose-dependent penetration of the Dleu2-17aa peptide, with maximal penetration at 25–50 µM (Fig. EV1B,C). We also analyzed the subcellular distribution of Dleu2-17aa by confocal microscopy and found that it was predominantly localized in the nucleus (Fig. EV1D). We hypothesized that this nuclear targeting was mediated by the high proportion (53%) of arginine residues in Dleu2-17aa, which resemble nuclear localization sequences (NLS) that facilitate protein import into the nucleus (Kalderon et al, 1984). Besides, cell proliferative assay showed that Dleu2-17aa had little effect on T cell proliferation (Fig. 2C).

We then differentiated naive CD4$^+$ and CD8$^+$ T cells into effector T (Teff) cells and Treg cells in the presence of Dleu2-17aa and its scrambled control peptide (scrPEP). Our results indicated that addition of exogenous Dleu2-17aa as well as overexpression of Dleu2-17aa promoted iTreg cell (CD4$^+$FoxP3$^+$) differentiation (Fig. 2D; Appendix Fig. S3), but does not significantly impact Tc1, Tc17, Th1, or Th17 subsets (Fig. EV2). Interestingly, we found that human DLEU2-25aa exhibited similar features in promoting mouse iTreg cell differentiation in vitro (Appendix Fig. S4). In addition, Treg suppression function assay revealed that Dleu2-17aa did not influence Treg cell suppression function (Fig. 2E). Collectively, these findings suggested that Dleu2-17aa may possess anti-inflammatory potential via promoting Treg cell differentiation.

## Dleu2-17aa interacts with Smad3 in T cells

To further clarify the mechanism of Dleu2-17aa in promoting Treg differentiation, we sought to identify its interaction target. We first detected the endogenous expression of Dleu2-17aa and found it to be mainly expressed in the nucleus (Fig. 3A). A pull-down assay was then performed using biotin-labeled Dleu2-17aa in EL4 cells and mouse CD4$^+$ T cells, followed by SDS-PAGE and liquid chromatography–mass spectrometry (LC-MS) analysis (Fig. 3B). Among the proteins enriched by LC-MS, we focused on those associated with Treg cell differentiation and function.

Our LC-MS analysis revealed several candidate interacting proteins (Fig. 3C and Dataset EV1). Among them, Smad3 was the most promising candidate interacting protein with high intensity in biotin-Dleu2-17aa samples but was not detected when large amounts of free Dleu2-17aa were competing with biotin-Dleu2-17aa (Fig. 3C). This interaction was verified using pull down

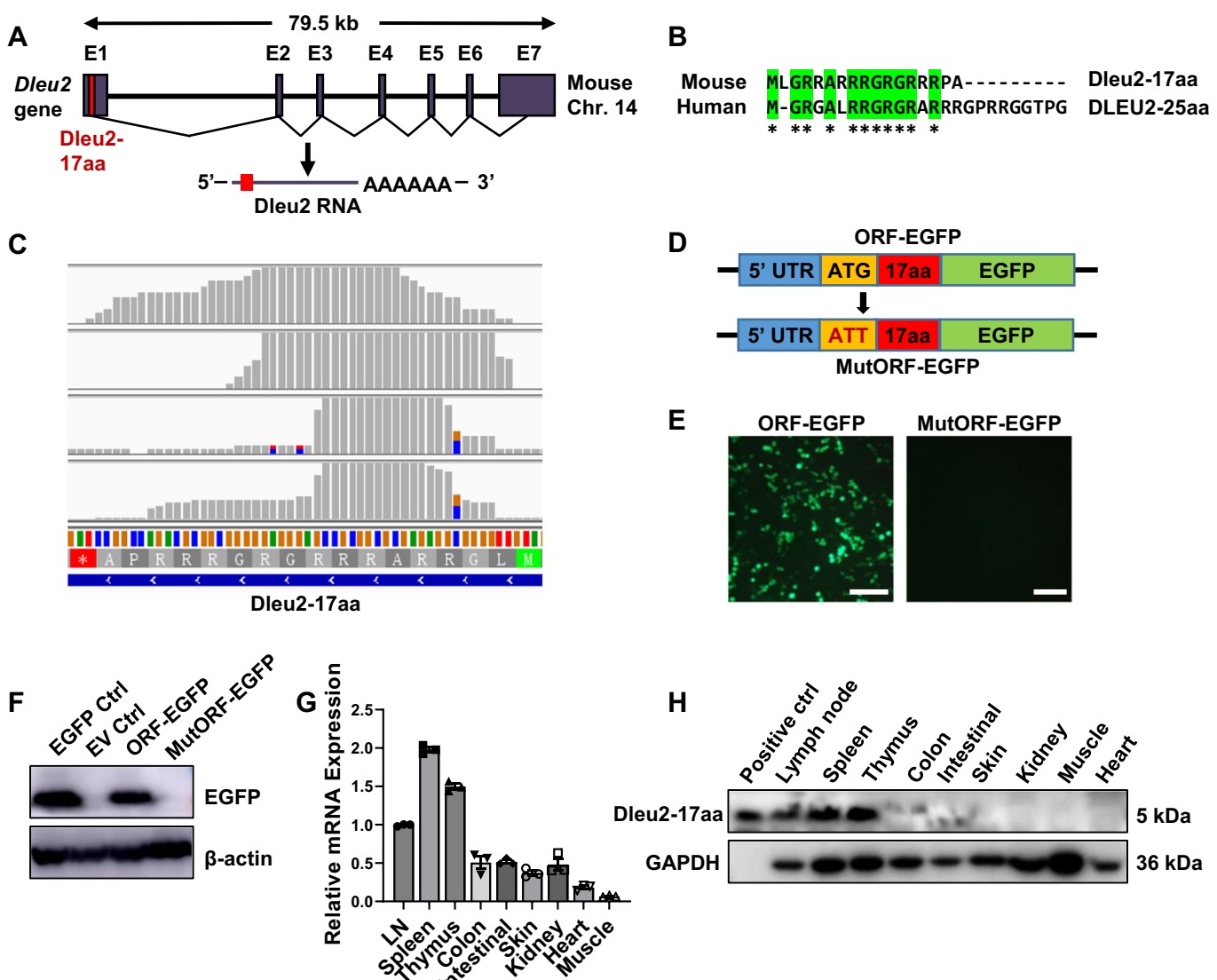

**Figure 1. Identification of Dleu2-17aa as an endogenously expressed micropeptide.**

(A) Schematic representation of the *Dleu2* gene locus with the predicted ORF indicated in red. (B) Sequence alignment of Dleu2-17aa and DLEU2-25aa. Identical amino acids are highlighted in green. (C) Ribo-seq reads mapping to the Dleu2-17aa ORF region in the mouse genome, visualized by the IGV browser. The data are from four Ribo-seq experiments (SRR8130804, SRR8130805, SRR7956051, SRR6189658). (D) Diagram of EGFP fusion constructs. The EGFP (without its own ATG) was inserted before the stop codon of Dleu2-17aa, then subcloned into the pcDNA3.1(-) vector. An ATT mutation was introduced to disrupt the Dleu2-17aa ORF to generate MutORF-EGFP construct. (E) Immunofluorescence detection of EGFP (green) expression in HEK293T cells transfected with indicated constructs ($n = 3$). Scale bars: 100 μm. (F) Immunoblotting of fusion protein levels using an anti-EGFP antibody ($n = 3$). (G) Relative mRNA levels of Dleu2 in indicated organs of WT mice were detected by quantitative PCR (qPCR) ($n = 3$). (H) Immunoblotting detection of endogenously expressed Dleu2-17aa in indicated WT mouse organs ($n = 3$). Data information: $n$ indicates biological replicate. Error bars are mean ± SEM. Source data are available online for this figure.

western blotting in EL4 cells (Fig. 3D). We next performed bio-layer interferometry (BLI) to quantify the interaction between Dleu2-17aa and Smad3. The results demonstrated a direct binding with high affinity between these two proteins, with an equilibrium Kd of 2.43 μM (Fig. 3E).

## Dleu2-17aa promotes Foxp3 transcription by modulating TGF-β/Smad signaling

Smad3 is a key transcription factor that mediates TGF-β-induced Treg differentiation (Schlenner et al, 2012; Tone et al, 2008). We observed

that the addition of Dleu2-17aa upregulated the expression of TGF-β-Smad signaling downstream genes and Treg cell-related genes (Fig. 4A and Appendix Fig. S5), suggesting that Dleu2-17aa promotes Treg differentiation through the TGF-β-Smad pathway. Upon TGF-β stimulation, the Smad complex (containing Smad3) was activated before it entered the nucleus and bound to the Smad binding site in the *Foxp3* CNS1 region (Schlenner et al, 2012; Tone et al, 2008). We found that the phosphorylation level of Smad3 remained unchanged in CD4[+] T cells upon Dleu2-17aa treatment (Fig. EV3A), suggesting that the modulation of Smad3 activity by Dleu2-17aa occurs after Smad3 activation.

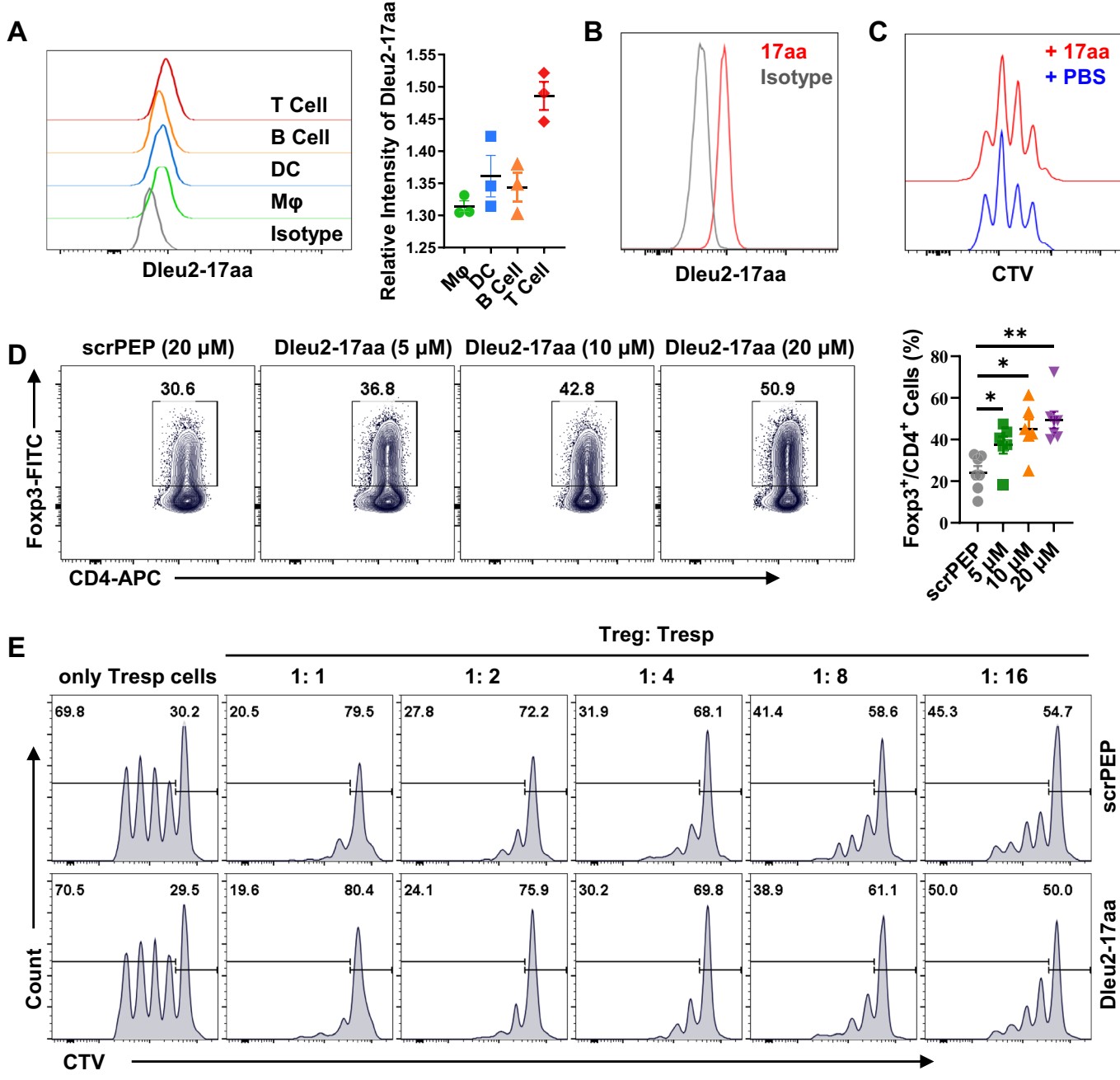

**Figure 2. Dleu2-17aa induces Treg cell differentiation in vitro.**

(A) Representative histogram (left panel) and relative median fluorescence intensity (MFI) (The MFI of each group was compared to the MFI of the isotype control) (right panel) of the Dleu2-17aa expression in various immune cell subsets from WT mice spleen ($n = 3$). (B) Representative flow histogram showing Dleu2-17aa expression in CD4$^+$ T cells ($n = 3$). (C) Representative flow cytometry of CTV histograms for CD4$^+$ T cells stimulated with anti-CD3 and anti-CD28 antibodies and cultured with either PBS or Dleu2-17aa (10 μM) for 3 days ($n = 3$). (D) Phenotype (left panel) and relative frequencies (right panel) of iTreg subsets in micropeptide-treated CD4$^+$ T cells of WT mice. Sorted naive CD4$^+$ T cells were incubated with scrPEP or Dleu2-17aa at indicated concentrations for 72 h under iTreg induction conditions and Foxp3 expression was then analyzed in live cells ($n = 6-7$). (E) CD4$^+$CD25$^+$ Treg cells and CD4$^+$CD25$^-$CD62L$^+$ naive T cells (Tresp) were sorted by FACS and co-cultured at decreasing ratios in the presence of anti-CD3 (1 μg/ml) plus irradiated Rag1$^{-/-}$ splenocytes as antigen-presenting cells ($5 \times 10^4$) with either scrPEP or Dleu2-17aa (10 μM) treatment. Treg suppressive function was assessed by Tresp cells proliferative assay using CTV staining ($n = 3$). Data information: $n$ indicates biological replicate. Error bars are mean ± SEM; *$P < 0.05$, **$P < 0.01$. Statistical analysis was by two-tailed Student's t-test for (D). Source data are available online for this figure.

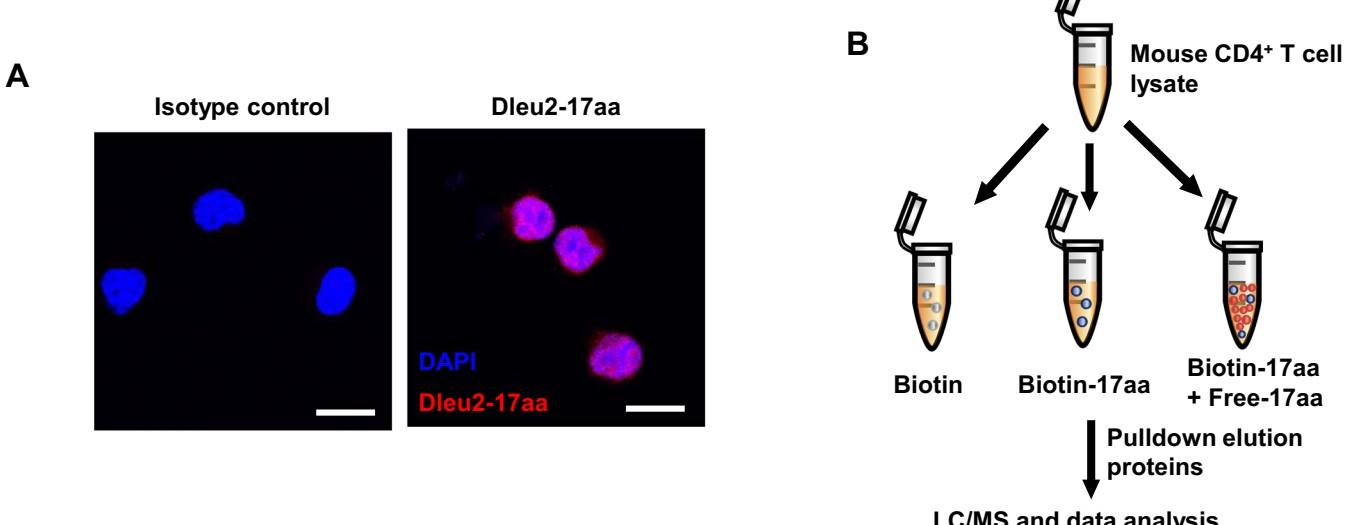

**C**

| Accession | Description | -10lgP | Biotin intensity | Biotin-17aa intensity | Biotin-17aa + Free-17aa intensity |
|---|---|---|---|---|---|
| sp\|Q8BUN5\|SMAD3_MOUSE | Smad3 | 108.44 | 0 | 1.47E+06 | 0 |
| tr\|A0A0A6YW89\|A0A0A6YW89_MOUSE | Pfkfb3 | 70.44 | 0 | 6.27E+06 | 0 |
| sp\|Q8VCF0\|MAVS_MOUSE | Mavs | 68 | 0 | 3.51E+05 | 0 |
| sp\|P13864\|DNMT1_MOUS | Dnmt1 | 35.92 | 0 | 1.44E+06 | 0 |

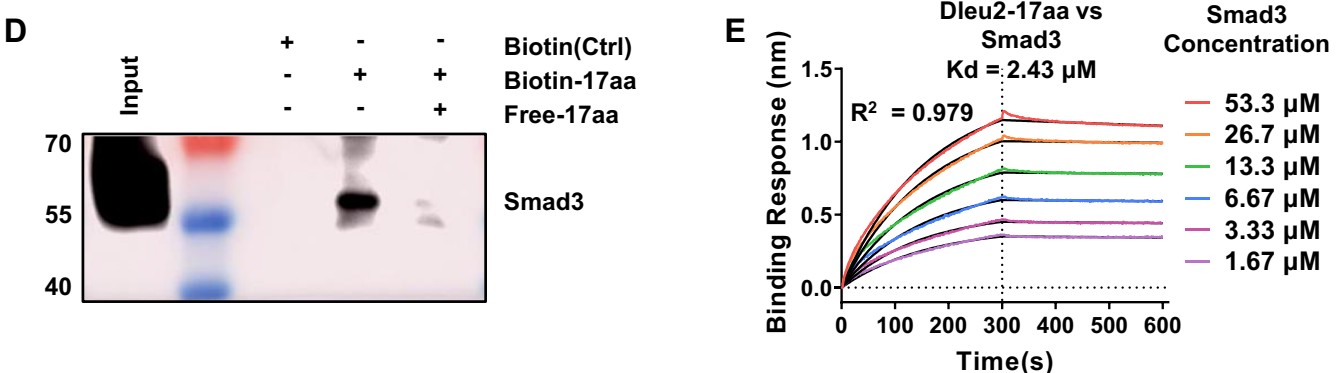

**Figure 3. Identification of Smad3 as the direct target of Dleu2-17aa.**

(A) Immunofluorescence imaging of endogenous Dleu2-17aa expression in CD4+ T cells ($n = 3$). Scale bars: 20 µm. (B) Schematic representation of the biotin pull-down assay to identify Dleu2-17aa interacting proteins in mouse CD4+ T cells. Biotinylated Dleu2-17aa was incubated with cell lysates, the interacting proteins were subjected to SDS-PAGE, excised and LC-MS identification. The specificity of the interaction was verified with excess free Dleu2-17aa to compete with the biotinylated form. (C) List of top-ranked and Treg cell-related proteins with highest intensity and significance score detected by MS analysis in (B). The significance score is calculated as the $-10\log10$ of the significance testing *p*-value. Paired t-test is used for significance calculation. (D) Immunoblotting verification of interaction between Dleu2-17aa and Smad3 ($n = 3$). (E) BLI sensor grams, colored according to Smad3 concentration (0 µM–53.3 µM), showed specific binding with Kd of 2.43 µM (data are average Kd values ± standard error calculated from linear regression curve fits, $n = 3$). Data information: n indicates biological replicate. Source data are available online for this figure.

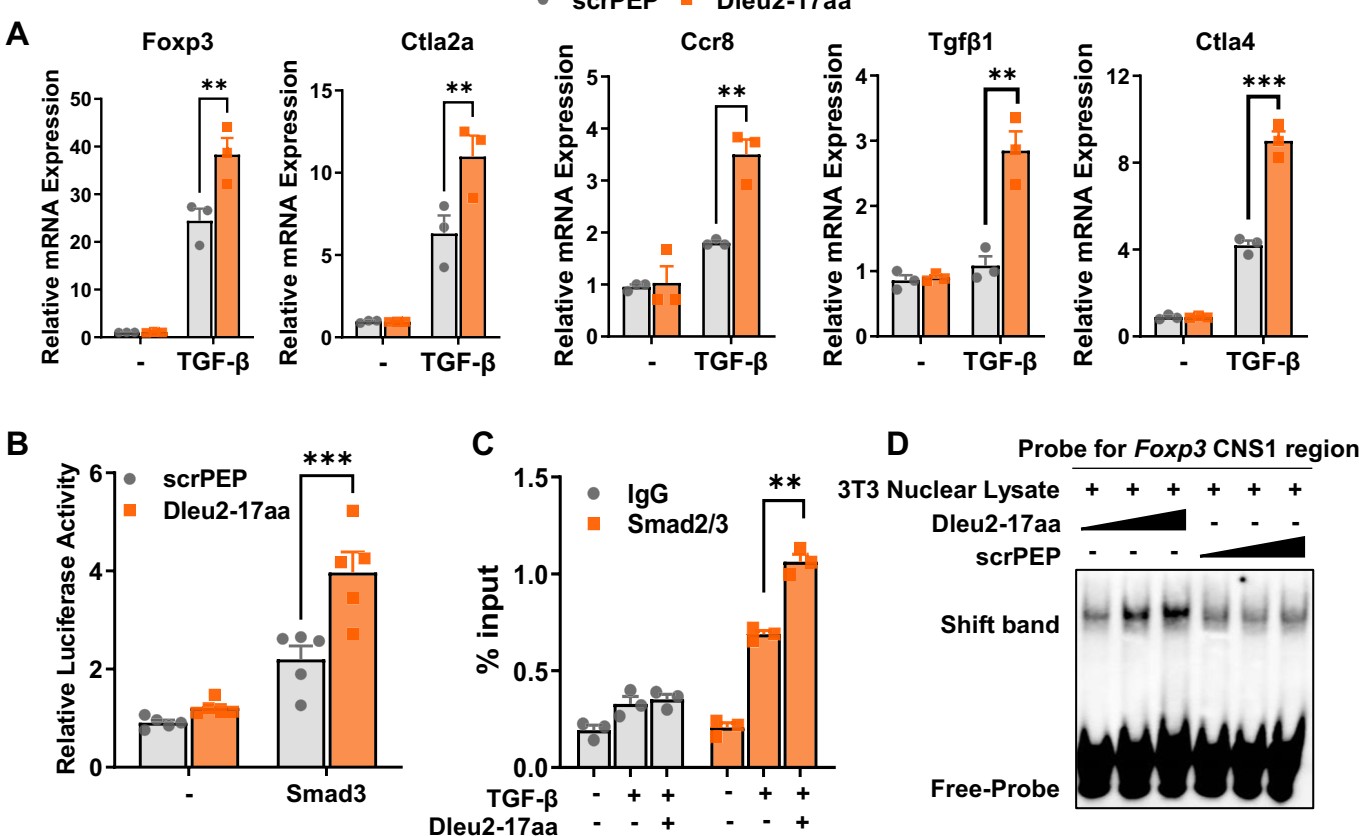

**Figure 4. Dleu2-17aa modulates TGF-β/Smad signaling to promote Foxp3 transcription.**

(A) qPCR analysis of TGF-β/Smad signaling downstream genes in CD4$^+$ T cells treated with Dleu2-17aa or scrPEP for 24 h and challenged with TGF-β to activate the signaling pathway ($n = 3$). (B) NIH/3T3 cells were transfected with a *Foxp3* CNS1-luciferase reporter plasmid and a Smad3 expression vector. Subsequently, the cells were treated with either Dleu2-17aa or scrPEP. Luciferase activity fold induction was quantified ($n = 3$). (C) *Foxp3* CNS1 region enrichment by Smad3 treated with or without Dleu2-17aa for 24 h in CD4$^+$ T cells was assessed by ChIP-qPCR ($n = 3$). (D) EMSA was performed in NIH/3T3 cell nuclear lysates using a 10 μM biotin-labeled *Foxp3* probe and treated with Dleu2-17aa or scrPEP (3, 10, 30 μM). There were band shifts to a higher molecular weight and an increase in band intensity by Dleu2-17aa treatment. Data information: n indicates biological replicate. Error bars are mean ± SEM; **$P < 0.01$, and ***$P < 0.001$. Statistical analysis was by two-tailed Student's t-test for (A–C). Source data are available online for this figure.

We hypothesized that Dleu2-17aa facilitates the binding of Smad3 to the *Foxp3* gene CNS1 region. We then performed luciferase reporter assays using a construct containing the *Foxp3* CNS1 region. NIH/3T3 cells were co-transfected with the CNS1-luciferase plasmid and a Smad3 expression vector. Addition of Dleu2-17aa resulted in a significant increase in *Foxp3* CNS1 region-dependent luciferase activity compared with scrPEP treatment (Fig. 4B), indicating that Dleu2-17aa promotes Smad3 transcription ability.

We also performed chromatin immunoprecipitation (ChIP) coupled with quantitative PCR (qPCR) analysis, the results showed that in activated CD4$^+$ T cells, Smad3 interacted with the *Foxp3* CNS1 region upon TGF-β stimulation, and Dleu2-17aa treatment further enhanced this interaction (Fig. 4C). The electrophoretic mobility shift assay (EMSA) also indicated that Dleu2-17aa strengthened Smad3 binding to the CNS1 region of *Foxp3* in a dose-dependent manner, while scrPEP did not (Figs. 4D and EV3B,C). Smad3 interacts with DNA through distinct binding elements that are present in many of its target genes (Derynck et al, 1998). We detected if Dleu2-17aa enhances Smad3 binding to three

of its common interaction sites, *Foxp3* CNS1 region (Tone et al, 2008), *AP-1* promoter region (Zhang et al, 1998) and *PAI-1* promoter region (Dennler et al, 1998). We found that Dleu2-17aa enhanced Smad3 binding to all three response elements tested (Fig. EV3D). Taken together, these results demonstrate Dleu2-17aa facilitates Smad3 binding to the *Foxp3* CNS1 region, thereby promoting Foxp3 transcriptional activation.

## Dleu2-17aa attenuates T-cell-mediated autoimmunity in mice

The distinct effect of Dleu2-17aa on upregulating Treg cell differentiation prompted us to evaluate its potential therapeutic effect in vivo. We employed experimental autoimmune encephalomyelitis (EAE), a well-established T-cell-mediated autoimmune disease model that resembles multiple sclerosis (MS) (Steinman and Zamvil, 2005; Pk and Rh, 1949). We immunized WT mice with myelin oligodendrocyte glycoprotein peptide fragment 35–55 (MOG$_{35-55}$) emulsified in complete Freund's adjuvant (CFA). Nine days post immunization of MOG, we administered Dleu2-17aa or

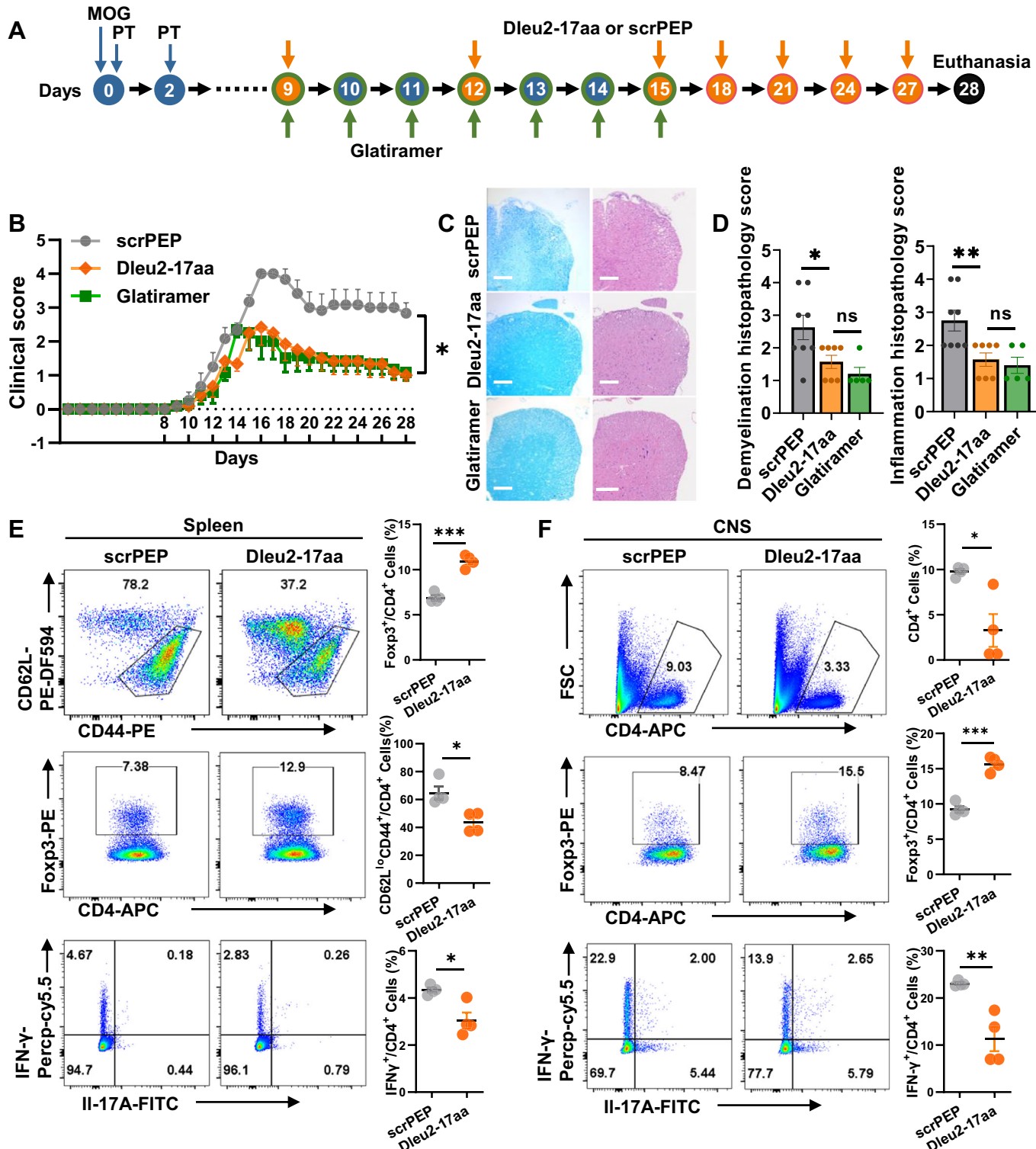

scrPEP intravenously every 3 days for a total of 7 injections. We used Glatiramer acetate (GA, i.e., Copaxone), a widely used drug in the treatment of MS and EAE, as a positive control and administered it subcutaneously daily for 7 times (Fig. 5A).

We found that Dleu2-17aa significantly reduced the clinical severity and the spinal cord demyelination of EAE mice,

comparable to Copaxone (Fig. 5B–D). On day 16 post-immunization, when disease severity reached its peak, we analyzed lymphocyte frequency in the central nervous system (CNS) and spleen. The results showed that Dleu2-17aa dampened inflammatory CD4+IL-17A+ T cell infiltration and increased CD4+Foxp3+ T cell accumulation in both organs compared with scrPEP-treated

**Figure 5.   Dleu2-17aa attenuates EAE by promoting Treg cells.**

(A) Schematic diagram of MOG-induced EAE and treatment strategies with Dleu2-17aa, scrPEP, and Glatiramer. Mice received 100 μg of Dleu2-17aa or scrPEP every 3 days via i.v. injection or 150 μg of Glatiramer every day for seven times via s.c. on days 9 to 15. (B) Clinical scores were examined daily from day 9 using a scoring system ranging from 0 (no sign) to 5 (complete paralysis or dead). The x-axis represents experimental days, and the y-axis represents symptom severity ($n = 8–9$). (C) Representative Luxol fast blue (LFB) and hematoxylin and eosin staining (H&E) staining sections for treated EAE spinal cord. Scale bars: 100 μm. (D) Histopathology scores of inflammations and demyelination in treated EAE mice spinal cord using a scoring system ranging from 0 (no inflammatory cells or demyelination) to 4 (extensive inflammatory cells infiltration or confluent foci of demyelination) ($n = 5–8$). (E) Representative dot plots (left panel) and frequency (right panel) of CD44$^{hi}$CD62L$^{lo}$ activated T cells, Foxp3$^+$ Treg cells, IL-17A$^+$ Th17 cells, and IFN-γ$^+$ Th1 cells gated on CD4$^+$ T cells in the spleen 14 days post immunization of MOG ($n = 3$). (F) Representative dot plots and frequency of CD4$^+$, Foxp3$^+$ Treg cells, and IFN-γ$^+$ Th1 cells gated on CD4$^+$ T cells in the CNS 14 days post immunization of MOG ($n = 3$). Data information: n indicates biological replicate. Error bars are mean ± SEM; *$P < 0.05$, **$P < 0.01$, and ***$P < 0.001$, n.s. indicates no significant difference. Statistical analysis was by two-way ANOVA for (B) and two-tailed Student's t-test for (D–F). Source data are available online for this figure.

controls (Fig. 5E,F). These results demonstrated that Dleu2-17aa promotes Treg differentiation in vivo and ameliorates T cell-mediated autoimmunity in EAE mice.

## Dleu2-17aa depletion reduces iTreg differentiation through dampened Smad3 binding to the *Foxp3* CNS1 region

To validate the function of endogenous Dleu2-17aa, we generated Dleu2-17aa knockout (KO) mice using CRISPR-Cas9 technology. We mutated the start codon ATG of Dleu2-17aa to ATT, which abolished its translation without affecting its host gene transcription (Fig. EV4A). We verified the successful knockout of Dleu2-17aa by sequencing, microscopy, and flow cytometry analyses (Figs. 6A and EV4B,C).

We measured the size of LN, SP, and Thy in KO mice, as well as the total cell number of SP and found no significant difference compared with WT mice (Fig. 6B,C). We also analyzed the development and distribution of T cell subsets in KO mice under steady-state conditions. We observed that the frequency and absolute number of T cell subsets in these organs were comparable between KO and WT mice (Fig. 6D; Appendix Fig. S6A). The T cell proliferation was also unaffected by the loss of Dleu2-17aa (Appendix Fig. S6B). These results indicated that Dleu2-17aa is dispensable for thymic selection and homeostatic maintenance of T cells in secondary lymphoid organs.

Next, we evaluated the role of endogenous Dleu2-17aa in iTreg differentiation using KO mice. We observed that Dleu2-17aa deficiency impaired iTreg differentiation in vitro (Fig. 6E,F). However, we did not detect any changes in the frequency of nTregs in LN, SP, and Thy of KO mice (Fig. 6G,H), nor any alteration in their suppressive function (Appendix Fig. S6C). In addition to iTregs, peripherally-derived Tregs (pTregs) can also be induced in tissues like the gut in response to commensal and dietary antigens. Smad3 binding the CNS1 region of *Foxp3* is critical for driving colonic pTreg differentiation (Atarashi et al, 2011; Lathrop et al, 2011; Schlenner et al, 2012). To investigate whether Dleu2-17aa regulates pTreg induction in vivo, we examined Foxp3$^+$ Tregs in WT and KO mice colonic lamina propria (cLP) under homeostatic conditions. We observed no difference compared with WT mice (Fig. 6I,J). Furthermore, we examined the effect of Dleu2-17aa on Smad3 binding to the *Foxp3* CNS1 region in response to TGF-β stimulation using KO mice CD4$^+$ T cells. Intriguingly, our data revealed that depletion of Dleu2-17aa resulted in reduced Smad3 binding to the *Foxp3* CNS1 region (Fig. 6K). Taken together, these findings provided evidence

that Dleu2-17aa plays a critical role in iTreg differentiation through facilitating Smad3 binding to the *Foxp3* CNS1 region.

## Deletion of Dleu2-17aa impairs autoimmune neuroinflammation

To investigate the role of endogenous Dleu2-17aa in regulating CD4$^+$ T cell-mediated immune responses in vivo, we induced active EAE in both WT and KO mice and compared their disease progression and severity. We found that Dleu2-17aa deficiency increased the susceptibility to EAE, as evidenced by higher clinical scores and more pronounced body weight loss in KO mice than in WT mice (Fig. 7A,B). Moreover, KO mice showed more extensive demyelination and inflammation in the CNS, indicating increased inflammatory damage (Fig. 7C,D).

We then analyzed the T cell subsets in the CNS of WT and KO mice after EAE induction to clarify the inflammatory responses. We found that the KO mice had a higher frequency of CD4$^+$ T cells and Th1 cells (Fig. 7E,F). In contrast, these mice had a lower frequency of Treg cells (Fig. 7G), which may suppress overreactive immune responses in the EAE mouse model. This imbalance resulted in a shift towards a more pro-inflammatory response which exacerbated the disease severity. To investigate if Dleu2-17aa affects CNS pathology in EAE, we performed immunostaining of spinal cord sections on EAE mice treated with Dleu2-17aa or scrPEP. The results revealed Dleu2-17aa is expressed in all the CD4$^+$ T cell, as expected. Furthermore, we also observed some colocalization of Dleu2-17aa with microglia, limited colocalization with astrocytes, and minimal colocalization with neurons (Appendix Fig. S7). There were no significant differences in astrocyte or microglia activation between Dleu2-17aa-treated and scrPEP-treated EAE mice, based on morphology. Neuronal staining was also unaffected by Dleu2-17aa, indicating no significant alterations in CNS inflammation. However, we did observe reduced CD4$^+$ T cell infiltration in the spinal cords of Dleu2-17aa-treated EAE mice compared with scrPEP controls (Appendix Fig. S7). These data implied the immuno-regulatory activities of Dleu2-17aa may through influencing CD4$^+$ T cell function.

To specifically assess the recall response of myelin-reactive T cells, we isolated splenic CD4$^+$ T cells from WT and KO EAE mice and re-stimulated them with MOG$_{33-35}$ peptide. Flow cytometry analysis showed similar frequencies of cytokines like IFN-γ and IL-17A expression and activation marker CD44 in CD4$^+$ T cells (Appendix Fig. S8A), also, there were comparable proliferation profiles between two groups (Appendix Fig. S8B). This indicated that Dleu2-17aa does not directly suppress

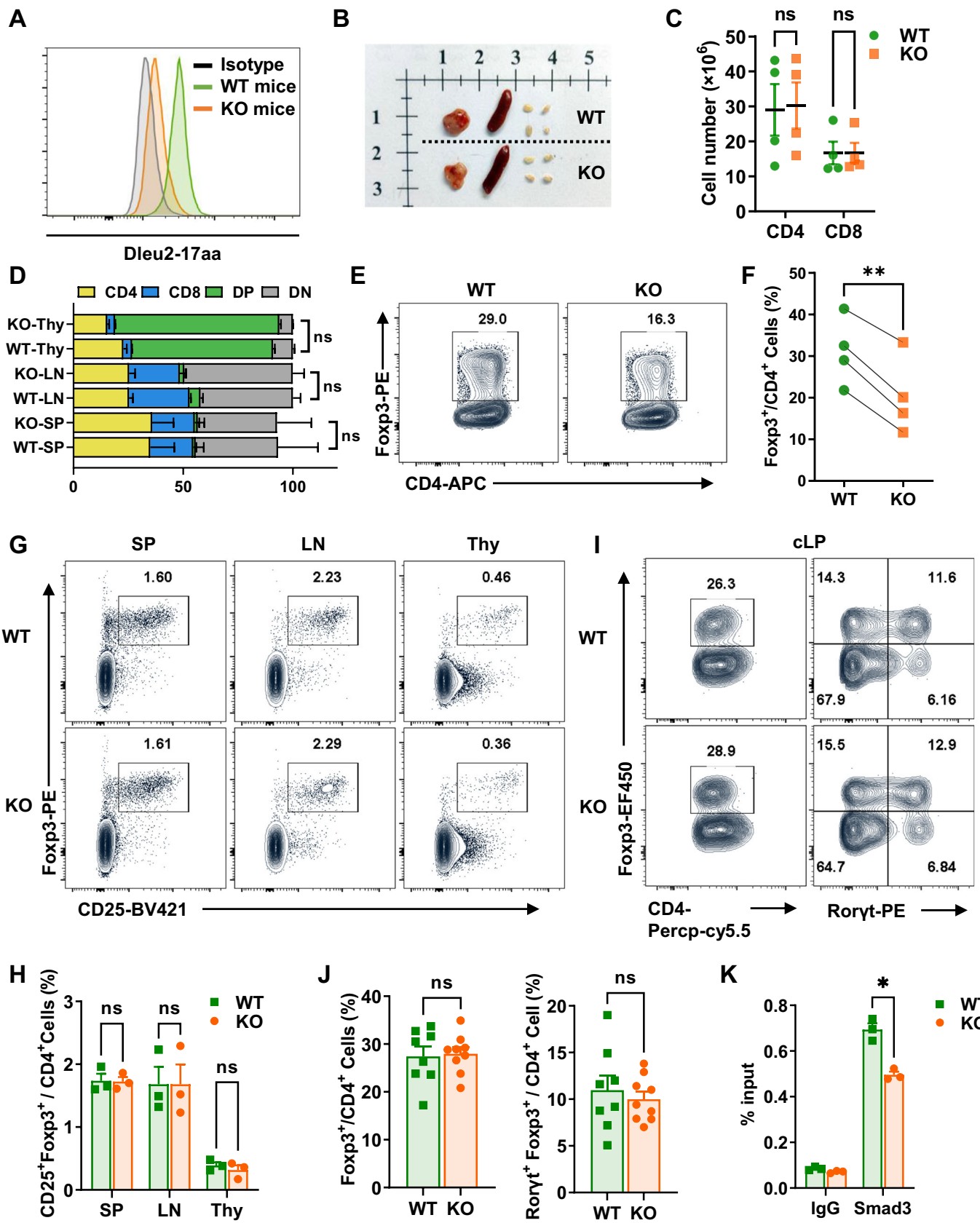

◀ **Figure 6.  The Role of endogenous Dleu2-17aa in T Cell development and differentiation.**

(A) Flow cytometry analysis of Dleu2-17aa expression in WT or KO mice spleen cells ($n = 3$). (B) Size of WT or KO mice Thy, SP, and LN (cm) ($n = 3$). (C) Total numbers of CD4$^+$ and CD8$^+$ T cells in WT and KO mice spleens ($n = 4$). (D) Relative percentages of CD4$^+$; CD8$^+$; CD4$^-$CD8$^-$ (DN); CD4$^+$CD8$^+$ (DP) cells in the LN, SP, and Thy ($n = 3$). (E,F). Phenotype (E) and relative frequencies (F) of iTreg subsets in CD4$^+$ T cells of WT or KO mice. Sorted naive CD4$^+$ T cells were under iTreg induction conditions for 72 h before analyzing Foxp3 expression in live cells ($n = 4$). (G) Representative dot plots of CD4$^+$CD25$^+$Foxp3$^+$ Tregs in the LN, SP, and Thy of WT and KO mice ($n = 3$). (H) Histograms showing the frequency of CD4$^+$CD25$^+$Foxp3$^+$ Tregs in the LN, SP, and Thy of WT and KO mice ($n = 3$). (I) Representative flow cytometry expression profiles of Foxp3 and RORγt from cLP of WT and KO mice ($n = 8$). (J) Histograms showing the frequency of CD4+Foxp3$^+$ and RORγt$^+$Foxp3$^+$ Tregs in the cLP ($n = 8$). (K) *Foxp3* CNS1 region enrichment by Smad3 in WT and KO mice CD4$^+$ T cells was assessed by ChIP-qPCR ($n = 3$). Data information: *n* indicates biological replicate. Error bars are mean ± SEM; *$P < 0.05$, **$P < 0.01$, n.s. indicates no significant difference. Statistical analysis was by two-tailed Student's t-test for (C,F,H,J,K). Source data are available online for this figure.

autoreactive T cell reactivation, but rather attenuates EAE by promoting Treg induction. Taken together, our results demonstrate that Dleu2-17aa is an important regulator of iTreg cell differentiation in vivo.

## Dleu2-17aa promotes dietary antigen induced pTreg differentiation in the colon

To examine if Dleu2-17aa regulates pTreg induction by dietary antigens, we utilized an ovalbumin (OVA) feeding model. Mice were gavaged with OVA and cholera toxin B (CTB) adjuvant every other day for 15 days to stimulate gut pTreg conversion. Both WT and KO mice did not show body weight loss over the 15-day period (Fig. EV5A). Flow cytometry analysis revealed significantly fewer Foxp3$^+$ pTreg cells especially Foxp3$^+$RORγt$^+$ pTregs in the cLP of OVA-fed KO mice compared with WT controls (Fig. EV5B–E). Given that the binding of Smad3 to *Foxp3* CNS1 region contributes to colonic pTreg induction, these data indicated Dleu2-17aa facilitates Smad3-dependent pTreg generation in the colon in response to fed antigens. This suggested a broad role for Dleu2-17aa in modulating both induced and peripheral Foxp3$^+$ Treg cell differentiation.

## Discussion

In this study, we identified and characterized an endogenous micropeptide encoded by the lncRNA *Dleu2* and revealed its role in regulating iTreg differentiation. *DLEU2* is the host gene for miR15a/16–1, it regulates B cell proliferation and has been reported to be deleted or epigenetically suppressed in leukemia (Mertens et al, 2006; Lerner et al, 2009; Klein et al, 2010). Previous studies have reported that DLEU2 is expressed in various cell types including immune cells such as B cells and T cells and mainly localized in the cytoplasm (He et al, 2021, 2; Li et al, 2022, 2; Dong et al, 2021), suggesting that it may have the potential to be translated into protein.

Recent advancements in deep-sequencing and bioinformatics technology have led to the discovery of micropeptides encoded by lncRNAs with distinct functions (Anderson et al, 2015; Bi et al, 2017; Ge et al, 2021; Wu et al, 2022). However, only a few micropeptides have been reported in immune cells. Our findings characterized the immunomodulatory function of a conserved 17-aa micropeptide encoded by the lncRNA *Dleu2* in CD4$^+$ T cells. By analyzing ribosome profiling data, we discovered a sORF is attached to ribosomes in mouse cells within the first exon of Dleu2, which indicates their translational potential. We found that

the 5′-terminal regions of *Dleu2* and *DLEU2*, which encode the micropeptide Dleu2-17aa and DLEU2-25aa, is conserved between mouse and human. Subsequent validation experiments confirmed that this 17aa-micropeptide is a new case of lncRNA-encoded peptide. Further investigation revealed that Dleu2-17aa could penetrate into CD4$^+$ T cells and induce the polarization of iTreg cells rather than Tc1, Tc17, Th1, and Th17 cells. We also generated Dleu2-17aa knockout mice and found that they have impaired iTreg differentiation while immune organ development and T cell distribution are normal. These results suggested that Dleu2-17aa regulates immune responses by modulating iTreg differentiation. Our study focused specifically on CD4+ T cells, while Dleu2-17aa is highly expressed in immune organs and may have other functions in other cells that warrant further exploration.

Treg cell deficiency is a hallmark of autoimmunity (Göschl et al, 2019; Dominguez-Villar and Hafler, 2018). In vivo, differentiated iTreg cells are stable and may provide long-lasting therapeutic effects for autoimmune diseases (Chen et al, 2011; Haribhai et al, 2016). In our study, we showed that Dleu2-17aa treatment ameliorated EAE, a mouse model of multiple sclerosis, by reducing the numbers of inflammatory/effector CD4$^+$ T cells and increasing the frequency of Treg cells in both the spleen and CNS of EAE mice. In contrast, the depletion of Dleu2-17aa exacerbated EAE severity as shown by our Dleu2-17aa knockout mice. Previous studies have shown that factors like retinoic acid and histone deacetylase inhibitors can induce or stabilize Foxp3 expression, promoting iTreg differentiation in vivo and treating autoimmune diseases (Xiao et al, 2008; Wang et al, 2009, 2015). Also, the adoptive transfer of Treg cells can modulate innate and adaptive immune responses and attenuate disease in animal models (Czaja, 2015; Haribhai et al, 2016; Selck and Dominguez-Villar, 2021; Mohammadi et al, 2021; Wu et al, 2021). Our findings suggested that Dleu2-17aa may act through a similar mechanism and provide clinical benefits as a supplement to existing approaches for treating autoimmune diseases. In addition to impaired iTreg formation, we found Dleu2-17aa KO mice exhibited reduced pTreg induction in the gut mucosa in response to feeding ovalbumin antigen. These demonstrate a broad role for Dleu2-17aa in modulating both induced and peripheral Foxp3$^+$ Treg cell differentiation.

Peptide drugs have become increasingly popular in the treatment of diseases due to their high safety and target affinity. With more than 100 peptide drugs currently on the market, these peptides or polypeptides (oligomers or short polymers of amino acids) have a wide range of therapeutic effects and are typically composed of 10-100 amino acids (Vasconcelos et al, 2013; Wang et al, 2022; Muttenthaler et al, 2021). They can serve as hormones,

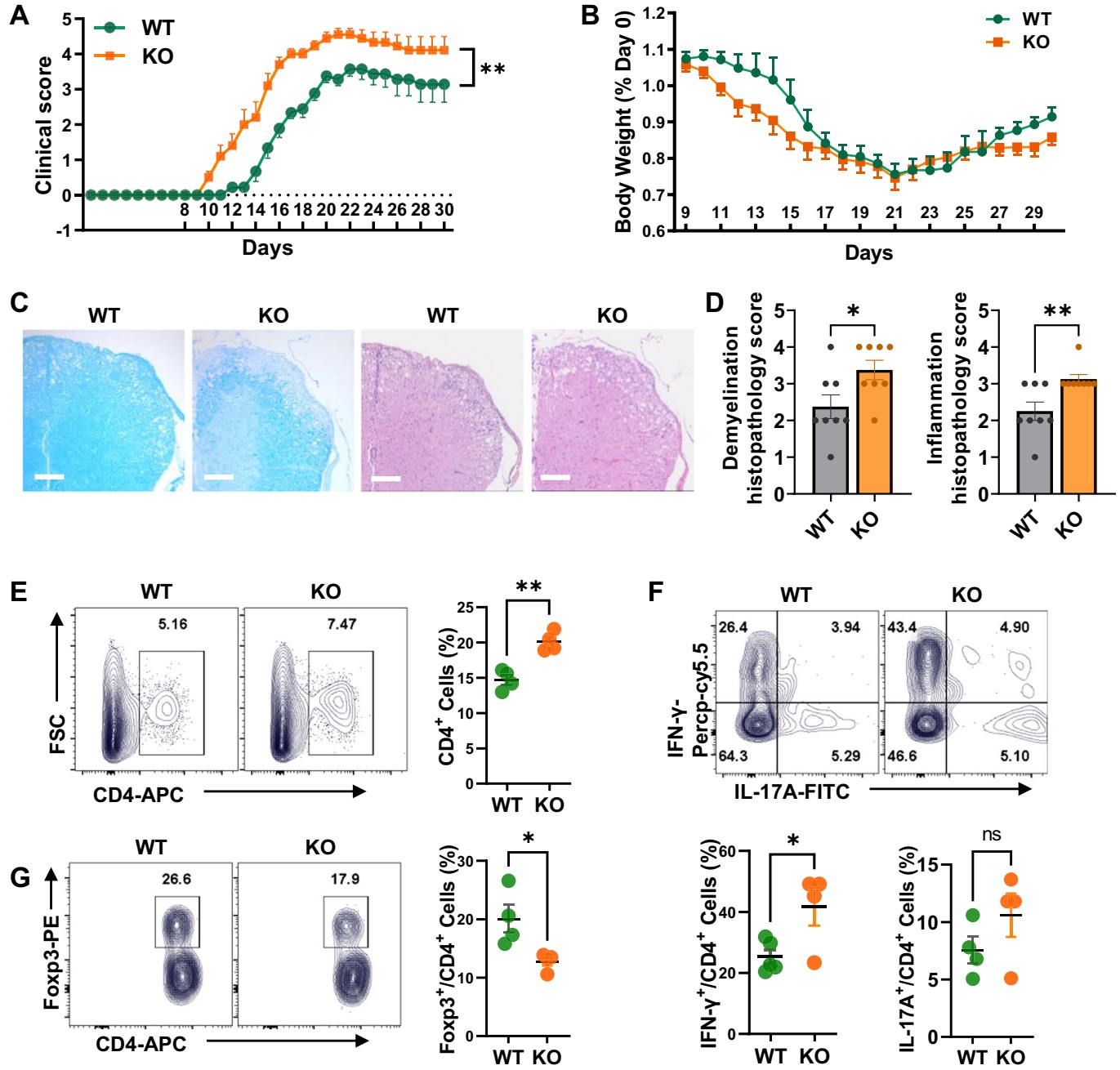

**Figure 7. Increased EAE severity in Dleu2-17aa knockout mice.**

(A) EAE clinical scores were examined daily from day 9 using a scoring system ranging from 0 (no sign) to 5 (complete paralysis or dead). The x-axis represents experimental days, and the y-axis represents symptom severity ($n = 6\text{–}8$). (B) Body weight was examined daily from day 9 post immunization of MOG ($n = 6\text{–}8$). (C) Representative LFB and H&E staining sections for treated EAE spinal cord. Scale bars: 100 μm. (D) Histopathology scores of demyelination (left panel) and inflammations (right panel) in WT or KO EAE mice spinal cord using a scoring system ranging from 0 (no inflammatory cells or demyelination) to 4 (confluent foci of demyelination) ($n = 8$). (E–G) Representative dot plots (left panel) and frequency (right panel) of CD4+ (E), IFN-γ+ Th1 cells, IL-17A+ Th17 cells (F) and Foxp3+ Treg cells (G) gated on CD4+ T cells in the CNS on day 14 post immunization of MOG ($n = 4$). Data information: n indicates biological replicate. Error bars are mean ± SEM; *$P < 0.05$, **$P < 0.01$, n.s. indicates no significant difference. Statistical analysis was by two-way ANOVA for (A) and two-tailed Student's t-test for (D–G). Source data are available online for this figure.

growth factors, neurotransmitters, ion channel ligands, and anti-infectives to treat conditions like type 2 diabetes, multiple sclerosis, high blood pressure and more (Vasconcelos et al, 2013; Wang et al, 2022; Muttenthaler et al, 2021). One class of peptide drugs that has

attracted biomedical interest is cell-penetrating peptides (CPPs). These peptides can cross cell membranes and deliver therapeutic agents directly to their targets. For example, tumor-targeting peptides have been used effectively for the delivery of

chemotherapeutic agents (Kang et al, 2020; Liu et al, 2017; Chen et al, 2021). Our study revealed that Dleu2-17aa is an immuno-modulatory CPP which induces iTreg differentiation and regulates immune responses. It may act as a molecular delivery vehicle by carrying other drugs into the cell nucleus where it interacts with its target protein, Smad3.

We identified Smad3 as the direct target of Dleu2-17aa. Smad3 is a crucial transcriptional factor to promote Foxp3 expression during Treg cell differentiation (Tone et al, 2008, 3; Gu et al, 2012, 3; Derynck and Zhang, 2003). In response to TGF-β stimulation, activated Smad2/3 recruits Smad4 and binds to the *Foxp3* CNS1 region (Tone et al, 2008, 3; Gu et al, 2012; Chen et al, 2003; Xu et al, 2010). Interestingly, we noted that Dleu2-17aa did not increase Smad3 phosphorylation. Instead, ChIP and EMSA assays revealed that Dleu2-17aa enhanced the binding of Smad3 to the *Foxp3* CNS1 region. The Smad-binding element (SBE) in the *Foxp3* CNS1 region acts as a sensor for TGF-β signaling. Mice deficient in *Foxp3* CNS1 region display iTreg deficiency and disrupted immune responses without changes in Th1 and Th17 cells (Josefowicz et al, 2012; Katoh et al, 2007, 9; Wu et al, 2014, 9). In line with its ability to enhance Smad3 binding to the *Foxp3* CNS1 region, exogenous Dleu2-17aa further upregulated downstream targets of TGF-β/Smad signaling genes. These results suggested that Dleu2-17aa acts as a regulator that promotes Foxp3 expression through modulation of the TGF-β/Smad signaling pathway.

In summary, Dleu2-17aa encoded by lncRNA *Dleu2* is highly expressed in immune-related organs and functions as a positive regulator of iTreg cell differentiation. Dleu2-17aa promotes Smad3 binding to the CNS1 region of *Foxp3*, regulates TGF-β/Smad signaling, and plays an anti-inflammatory role in autoimmune disease (Synopsis Fig). Our findings offered a novel mechanism in the Treg cell polarization process, expanded the function of Dleu2 in immunity, and highlight the therapeutic potential of Dleu2-17aa in autoimmune disease.

# Methods

## Mice

C57BL/6 (wild-type) mice at 6–12 weeks were purchased from Shanghai Laboratory Animal Center. All mice were housed under specific pathogen-free (SPF) conditions. Mice were sex and age-matched and randomly used for experiments. All mice were randomly used for all experiments in compliance with the National Institutes of Health Guide for the Care and Use of Laboratory Animals with the approval (SYXK-2003-0026) of the Scientific Investigation Board of Shanghai Jiao Tong University School of Medicine, Shanghai, China.

## Cell lines

All cell lines were obtained from American Type Culture Collection (ATCC) and cultured as suggested by the supplier except when otherwise notified. The mouse T cell line EL4 (TIB-39) cells and fibroblast cell line NIH/3T3 (CRL-1658) cells were cultured in high-glucose DMEM (Thermo Fisher Scientific) supplemented with 10% fetal bovine serum (FBS; Gibco). The human T cell line Jurkat (TIB-152) cells were cultured in RPMI 1640 medium (Thermo

Fisher Scientific) supplemented with 10% FBS. All lines were maintained at 37 °C in a humidified 5% $CO_2$ atmosphere.

## Generation of Dleu2-17aa KO mice

We used CRISPR/Cas9 technology to introduce a desired point mutation in mice through homologous recombination repair. Cas9 mRNA and guide RNA were obtained through in vitro transcription, with the gRNA sequence being GGTTCCCTGGTCCCCGATGT. Oligo donor DNA was synthesized with the mutated base (GAGCTTTGCTGAAACTGCACAAAAAATCGAGCTGGGGGG TTCCCTGGTCCCCGATTTTGGGGCGGAGAGCGCGGCGCCGA GGGAGGGGGCGGCGCAGACCGGCCTAGGGGA). Cas9 mRNA, gRNA and donor DNA was microinjected into fertilized eggs of C57BL/6J mice to obtain F0 generation mice. The genomic region surrounding the targeted locus was amplified by PCR using primers 5′-CTGGGGCTTTTCTCTCCTGG-3′ and 5′-GCTGCTGCTCC TCGTTCTAT-3′ and sequenced using forward primer 5′-CTGGGGCTTTTCTCTCCTGG-3′.

## Peptide sequence

Dleu2-17aa: MLGRRARRRGRGRRRPA; scrambled Dleu2-17aa: ARGPARRMRGGRRRRLR; and DLEU2-25aa: MGRGALRRGR GRARRRGPRRGGTPG.

## Plasmid constructs

Plasmids for bacterial expression and transient mammalian cell expression were in a pcDNA3.1(-) backbone (T7 and CMV promoters; Promega). For purification from bacterial overexpression, Smad3 cDNA sequences (RefSeq NM_016769.4) were subcloned to pET28a (Promega). To construct plasmids for dual reporter luciferase assays, Foxp3 promoter sequences (CAGAbox) were amplified and cloned into the pGL3 basic vector to become pGL3-CAGA.

## Experimental autoimmune encephalomyelitis (EAE) induction and treatment

For active EAE immunization, myelin oligodendrocyte glycoprotein peptide (MOGp35–55, MEVGWYRSPFSRVVHLYRNGK, Gen-Script) was emulsified in complete Freund adjuvant (Sigma-Aldrich). An equivalent of 300 μg MOG peptide was injected per mouse subcutaneously followed by 200 ng of pertussis toxin (Merck-Calbiochem) injection intravenously on day 0 and day 2 of immunization. We weighed mice daily and scored their disease Phenotype according to a standard EAE scoring scale: 0, no detectable symptoms; 0.5, partial tail weakness; 1, tail paralysis; 1.5, gait instability or impaired righting ability; 2, hind limb paresis or partial paralysis; 2.5, hind limb paralysis; 3, full hind limb paralysis with partial forelimb paresis or paralysis; 4, hind limb and forelimb paralysis; 5, moribund or death. Every 3 days (days 9, 12, 15, 18, 21, and 27), 100 μg of Dleu2-17aa or scrPEP was injected intravenously or 150 μg of Glatiramer was injected subcutaneously daily for 7 times into each mouse. Inflammation and demyelination scores were calculated. Briefly, spinal cords stained with H&E were used to score inflammation (0, no inflammatory cells; 1, a few scattered inflammatory cells; 2, small organization of inflammatory

infiltrates; 3, large organization of inflammatory infiltrates; and 4, extensive inflammatory cells infiltration). Spinal cords stained with luxol fast blue were used for demyelination scoring (0, no demyelination; 1, a few scattered naked axons; 2, small groups of naked axons; 3, large groups of naked axons; and 4, confluent foci of demyelination).

## Western blotting and antibodies

Cells and mouse tissues were lysed in RIPA lysis buffer (Beyotime) supplemented with protease and phosphatase inhibitor cocktails (Thermo Scientific). Cellular extracts were then prepared by sonication. Equal amounts of protein were loaded and separated using sodium dodecyl sulfate-polyacrylamide gel electrophoresis (SDS-PAGE). After transfer to a polyvinylidene difluoride (PVDF) membrane, the proteins were blocked with 5% bovine serum albumin (BSA) and incubated in primary antibodies at 4 °C overnight. Subsequently, horseradish peroxidase (HRP)-conjugated anti-rabbit or anti-mouse IgG (H + L) secondary antibody (1:1000; Beyotime) was used to incubate the samples for 1 h at room temperature. Primary antibodies of EGFP (1:10,000, Proteintech), Smad3 (1:1000, CST), Smad2/3 (1:1000, CST), β-actin (1:20,000, Proteintech) and GAPDH (1:20,000, Proteintech) were used. For Dleu2-17aa detection, lysis was conducted using Tricine–SDS-PAGE and transferred to a PVDF membrane at 150 mA for 30 min and then fixed with 1% glutaraldehyde solution (Servicebio) for 1 h. An anti-Dleu2-17aa antibody was used (1:400; HuaBio), and all other steps were the same as described above.

## Pull-down assays

For each sample, 1 million Mouse CD4$^+$ T cells or EL4 cells were collected and sonicated then lysed on ice with RIPA lysis buffer (Beyotime) with a protease and phosphatase inhibitor cocktail (1:100; Thermo Fisher Scientific) for 20 min and centrifuged for another 20 min at 12,000 × g at 4 °C, and cleared lysates were harvested. Samples were incubated with 2 μM of Biotin, Biotin-Dleu2-17aa, or Biotin-Dleu2-17aa and free Dleu2-17aa for 2 h at 4 °C with rotation. The streptavidin-agarose from Streptomyces avidin (20 μl per sample; Sigma-Aldrich) was washed six times with 0.1% Triton TBS and coincubated with protein lysates for 4 h at 4 °C with rotation. The coincubated beads were washed six times with lysis buffer and resuspended with 20 μl of lysis buffer and 5 μl of loading buffer. The mixture was boiled for 5 min at 100 °C and harvested by centrifugation. To detect the interaction protein with Biotin-Dleu2-17aa, samples were separated by SDS-PAGE, and visualized by silver staining. The protein-containing bands in the gel were excised, followed by in-gel digestion and analysis by LC-MS/MS. To detect the interaction protein with Biotin-Dleu2-17aa, samples were separated by SDS-PAGE, and visualized by western blot.

## Silver staining and LC-MS

The samples of immunoprecipitants pulled down by streptavidin-agarose from Biotin-Dleu2-17aa in mouse CD4$^+$ T cells or EL4 cells were separated with SDS-PAGE and stained with Pierce Silver Stain Kit (Thermo Fisher Scientific) according to the instruction manual. The protein-containing bands were cut for LC-MS. All MS was performed on a QE-Plus mass spectrometer connected to an Easy-nLC2000 via an Easy Spray (Thermo Fisher Scientific). Spectra were analyzed using PEAKS 8.0 (Bioinformatics Solutions).

## BLI binding assays

BLI assays of Smad3-Dleu2-17aa interaction were performed using an Octet RED96 instrument (ForteBio). 50 μg/ml of Biotin-labeled Dleu2-17aa was immobilized on Streptavidin-coated biosensors in MAT assay buffer containing 100 mM Tris-HCl pH 8.0, 20 mM MgCl$_2$ and 200 mM KCl. BSA was added to the assay buffer with a final concentration of 0.1 mg/mL to reduce non-specific interaction. The association of free Smad3 with Dleu2-17aa was measured at a range of protein concentrations (1.67, 3.33, 6.67, 13.3, 26.7, and 53.3 μM) for 800 s, followed by dissociation in DPBS (0.5% BSA, w/w) for 180 s. All experiments were performed at RT. Binding affinity (KD) was calculated based on two independent experiments using BLItz system software (ForteBio).

## T cell proliferation analysis

CD4$^+$ or CD8$^+$ T Cells were sort-purified and labeled with CellTrace Violet (CTV) according to the manufacturer's protocol (Invitrogen). 0.1 million CTV labeled T cells were cultured with plate-coated anti-CD3 (2.5 μg/ml; BD Biosciences) and soluble anti-CD28 (2 μg/ml; BD Biosciences) antibodies, in certain groups of cells were cultured in the present of Dleu2-17aa (10 μM) or PBS control the phenotype of proliferation was determined by flow cytometry.

## In vitro mouse T cell differentiation

C57BL/6 mice were sacrificed, and their spleens and peripheral lymph nodes were removed and gently dissociated into single-cell suspensions. Naive T cells were isolated using the Mouse CD4 Naive T Cell Isolation Kit (BioLegend, Stem Cell, or Thermo Fisher Scientific). Flow cytometry verified that the purity (CD4$^+$CD25$^-$CD62L$^{hi}$) of the enriched subpopulation was typically higher than 95%. All T cells were cultured using RPMI-1640 medium (Gibco) supplemented with 10% heat-inactivated fetal bovine serum (Gibco), 2 mM L-glutamine (Gibco), 10,000 units/mL of penicillin, 10,000 μg/mL of streptomycin, and 25 μg/mL of Gibco Amphotericin B (Thermo Fisher Scientific), 10 mM HEPES (Gibco), 1 mM Sodium pyruvate, MEM Non-Essential Amino Acids (Life Technologies) and 55 μM β-mercaptoethanol (Gibco). 0.1 million naive CD4$^+$ T cells were activated with plate-coated anti-CD3 (2.5 μg/ml; BD Biosciences) and soluble anti-CD28 (2 μg/ml; BD Biosciences) antibodies. The Th1 cell differentiation condition included recombinant murine (rm)IL-12 (10 ng/ml, PeproTech) and anti-IL-4 (10 μg/ml, BioLegend). The Th17 cell differentiation condition included rmIL-6 (10 ng/ml, PeproTech), rmTGF-β1 (5 ng/ml, PeproTech), rmIL-23 (10 ng/ml, PeproTech), anti-IL-4 (5 μg/ml, BioLegend) and anti-IFN-γ (10 μg/ml, eBioscience). The inducible T regulatory cell (iTreg) differentiation condition included rmTGF-β (5 ng/ml, PeproTech). 0.1 million naive CD8$^+$ T cells were activated with plate-coated anti-CD3 (10 μg/ml; BD Biosciences) and plate-coated anti-CD28 (10 μg/ml; BD Biosciences) antibodies. The Tc17 cell differentiation condition included rmIL-6 (10 ng/ml, PeproTech), rmTGF-β1 (2 ng/ml,

PeproTech), rmIL-23 (25 ng/ml, PeproTech), rmIL1-β (5 ng/ml, PeproTech), and anti-IFN-γ (10 µg/ml, eBioscience). The Tc1 cell differentiation condition included rmIL-2 (5 ng/ml, PeproTech) and rm IL-12 (1 ng/ml, PeproTech). Different concentrations of Dleu2-17aa and scrPEP (Genscript) were added to the T cell culture from the beginning unless specified. Cells were cultured in U-bottom 96-well plates for 72 h and detected by flow cytometry.

## In vitro Treg suppression assay

Tregs were sort-purified as DAPI⁻CD4⁺CD3⁺CD25^high T cells, and DAPI⁻CD4⁺CD25⁻CD44⁻CD62L⁺ T cells are sort-purified as activated responder T cells (Tresp). Splenocytes from Rag1^-/- mice were irradiated with 30 Gy of X-ray. Tresp cells ($5 \times 10^4$) were then labeled with Cell Trace Violet (CTV) according to the manufacturer's protocol (Invitrogen). Labeled Tresp were cultured with Treg cells at ratios of 1:1, 1:2, 1:4, 1:8, and 1:16 for 72 h in the presence of anti-CD3 (UCHT1, BioLegend). Dleu2-17aa or scrPEP were added in the concentration of 10 µM. The suppression function of Treg cells was analyzed by determining the dilution of the CTV label in responder cell proliferation.

## T cell recall assay

Naive CD4⁺ T cells were isolated from spleens of WT and KO mice at 9 days post-EAE immunization using magnetic separation. T cells were labeled with CellTrace Violet (CTV) dye. Irradiated antigen-presenting cells (APCs) were prepared from spleens of Rag1^-/- mice. CTV-labeled WT or KO CD4⁺ T cells ($1 \times 10^6$) were cocultured with Rag1^-/- APCs ($5 \times 10^6$) in the presence of MOG35-55 peptide (50 µg/ml). After 72 h, cells were stimulated with PMA/ionomycin for 5 h. Cytokine production was assessed by intracellular staining and flow cytometry. Proliferation was measured by CTV dilution.

## Flow cytometry

For in vitro mouse T cell differentiation, after 72 h of induction, cells were activated for 4–6 h at 37 °C with the Cell Stimulation Cocktail (Thermo Fisher Scientific) if intracellular cytokine staining is needed, surface markers were stained in FACS buffer (PBS containing 2% FBS) for 30 min at 4 °C protected from light. Cells were then fixed and permeabilized with the Foxp3 Staining Buffer Set (eBioscience) or BD Cytofix/Cytoperm (BD Biosciences) and were stained with fluorescent antibodies. After washing, stained cells were assayed with an LSRFortessa (BD) flow cytometer, and data were analyzed with FlowJo software. Single-cell suspensions from EAE mouse spleen or CNS (day 16 post-immunization) were derived through mechanical separation and passage through a 70-µm filter (Fisherbrand) and analyzed using similar steps. For flow cytometry, antibodies against CD4 (clone GK1.5, 1:200), CD62L (clone MEL-14, 1:200), CD44 (clone IM7, 1:200), CD25 (clone PC61.5, 1:200), IL-17A (clone eBio17B7, 1:200), IFN-γ (clone XMG1.2, 1:200), FoxP3 (clone FJK-16s, 1:200), CD184 (clone 2B11/CXCR4, 1:100), CD152 (clone BNI3 1:100) and CD357 (clone DTA-1 1:100) were from eBioscience, CD3 (clone 145-2C11, 1:200) was from Biolegend and TGFBI (1:100) were from proteintech. For Dleu2-17aa expression detection, C57BL/6 mice were sacrificed, and their spleens were removed and gently dissociated into single-

cell suspensions. An Fc blocker was used to reduce potential nonspecific antibody staining caused by IgG. Surface staining was performed using monoclonal antibodies against CD45 (clone 30-F11), CD3 (clone 145-2C11), CD19 (clone SJ25C1), CD11c (clone N418), MHCII (clone M5/114.15.2), F4/80 (clone BM8) all at a dilution of 1:200 (from Biolengend) followed by fixation and permeabilization of the nuclear membrane BD Cytofix/Cytoperm (BD Biosciences). Intracellular staining was performed using the Dleu2-17aa antibody at a dilution of 1:200 while normal rabbit IgG was used as an isotype control at the same concentration. After secondary antibody staining, cells were analyzed by flow cytometry.

## OVA oral administration and colon lamina propria lymphocyte Isolation

WT and KO mice were orally gavaged with OVA protein (300 µg) together with CTB adjuvant (10 µg) every other day for 15 days. Colons were harvested and lamina propria lymphocytes isolated. Cells were surface stained with anti-CD4, then fixed and permeabilized for intracellular staining with anti-Foxp3 and anti-RORγt antibodies, followed by flow cytometry analysis.

## Immunofluorescence microscopy

Mouse CD4⁺ T cells and EL4 cells were seeded on glass coverslips and incubated with FAM-Dleu2-17aa (10 µM) for 4 h. Cultured cells were washed three times in PBS and fixed with 4% PFA for 10 min at RT, then incubated in Blocking Buffer (1 × PBS/5% BSA/0.3% Triton X-100) for 1 h. After being washed in PBS and air drying, the coverslips were mounted with DAPI (1:1000; Thermo Fisher Scientific). Images were acquired with Leica TCS SP8 laser scanning confocal microscope.

## Luciferase reporter assay

NIH/3T3 cells were seeded at $5 \times 10^5$ cells/well in 24-well plates and transfected with 0.2 µg/well of pGL3 *Foxp3* CNS1-luciferase reporter and the internal control vector pRL-TK (Promega) at a ratio of 10:1 (reporter construct: control vector) using LipofectamineTM 2000 according to the manufacturer's protocol; For specific groups, co-transfect with pcDNA3.1-Smad3, an empty vector served as the control. After 7 h, remove the transfection medium and replace it with fresh medium containing either 20 µM Dleu2-17aa or PBS. 24 h post-transfection, luciferase activity was measured using the Dual-Luciferase® Reporter Assay System (Promega) according to the manufacturer's protocol.

## ChIP-qPCR

Mouse CD4⁺ T cells were incubated with Dleu2-17aa for 24 h, and TGF-β activated for 2 h. ChIP assays were performed using the SimpleCHIP enzymatic chromatin immunoprecipitation kit (Cell Signaling Technology, #9002) according to the manufacturer's protocol with minor modifications. In brief, cells were harvested and cross-linked in 1% (v/v) formaldehyde and then lysed for 10 min at RT. Chromatin was sheared by sonication. Subsequently, nuclei were isolated by the lysis of cytoplasmic fraction. Chromatin was digested into fragments of 150–900 bp by micrococcal nuclease (400 gel units) for 20 min at 37 °C, followed by ultrasonic

disruption of the nuclear membrane using a standard microtip and a Branson W250D Sonifier (four pulses, 60% amplitude, and duty cycle 40%). The input group accounted for 1% of the total DNA. The chromatins were immunoprecipitated overnight at 4 °C with 5 μg of either anti-Smad2/3 (CST) Ab or the negative control IgG. After incubation with 30 μl of ChIP grade protein G-agarose beads for 2 h at 4 °C, the Ab-protein-DNA complexes were then eluted from the beads and digested by Proteinase K (40 μg) for 2 h at 65 °C, followed by spin column-based purification of the DNA. Finally, genomic DNA recovered from the ChIP assays was qPCR amplified with primers specific to the Smad3-binding elements of the *Foxp3* CNS1 region. The primers used for the detection of the *Foxp3* binding sequence were as follows: forward, 5′-CCCATGTTGGCTTCCAGTCTCCTTTATGG-3′ and reverse, 5′-ACCCAGGCTCTTAACCTCTCTGTACCT-3′. The signals were expressed as a percentage of the total input chromatin.

## Electrophoretic mobility shift assay

EMSA was performed using the LightShift Chemiluminescent EMSA kit (Thermo Fisher Scientific, #20148). Briefly, Nuclear extracts of NIH/3T3 cells were prepared using theNE-PER Nuclear and Cytoplasmic Extraction Kit (Thermo Fisher Scientific, #78835). Complementary DNA oligonucleotide *Foxp3* CNS1 region probes were 5′-TATGGGAGCCAGACTGTCTGGAACAACCT-3′ and 5′-AGGTTGTTCCAGACAGTCTGGCTCCCATA-3′, AP-1 probes were 5′-CTAGCTCTCTGACGTCAGGCAATCTCT-3′ and 5′-AGAGATTGCCTGACGTCAGAGAGCTAG-3′, PAI-1 probes were 5′-TCGAGAGCCAGACAAGGAGCCAGACAAGGAGCCA GACAC-3′ and 5′-GTGTCTGGCTCCTTGTCTGGCTCCTTGT CTGGCTCTCGA-3′. Probes were end-labeled with biotin and annealed. The reaction mix consists of labeled oligonucleotides, 10 μg of the nuclear extracts, and 3, 10, 30 μM Dleu2-17aa or scrPEP. The binding reaction was incubated at room temperature for 30 min, followed by loading onto a 6% native DNA polyacrylamide gel. The gel was prerun and run with 0.5% Tris-Borate-EDTA and processed according to the manufacturer's instructions.

## mRNA extraction and quantitative RT-PCR

Total RNA from cells or tissues was extracted using TRIzol reagent (Invitrogen) and then quantified. 1 μg of total RNA was reverse transcribed into cDNA with HiScript II Q RT SuperMix for qPCR (+gDNA Wiper) (Vazyme, China, R223-01). The cDNA samples were run in triplicate at 200 ng/well. Quantitative real-time PCR (qPCR) was performed using ChamQ Universal SYBR qPCR Master Mix (Vazyme, China) on an Applied Biosystems 7500 Fast Real-Time PCR System or a ViiA 7 Real-Time PCR System (Applied Biosystems). All gene expression results were normalized to the expression of the housekeeping gene GAPDH. Primer sequences are listed as follows: Dleu2 forward, 5′-GTAATG-CATTGGAATATGATAGGCG-3 and reverse, 5′-GCACATCTTT-CAAAGCCAAATCC-3; *Foxp3* forward, 5′-CCCATCC CCAGGAGTCTTG-3′ and reverse, 5′-ACCATGACTAGGGG-CACTGTA-3′; *Ctla2a* forward, 5′-CTGCTTGGGAATGATGT-CAGCTG-3′ and reverse, 5′-TTCTCCTCCCACACGAGTCTTC-3′; *Ccr8* forward, 5′-CTGCGATGTGTAAGGTGGTCTC-3′ and reverse, 5′-CCTCACCTTGATGGCATAGACAG-3′; *Tgf-β* forward:

5′-TGATACGCCTGAGTGGCTGTCT-3′ and reverse, 5′-CACAA-GAGCAGTGAGCGCTGAA-3′; *Ctla4* forward, 5′-ACTCATG-TACCCACCGCCATA-3′ and reverse, 5′-GGGCATGGTTC TGGATCAAT-3′; *Gapdh* forward, 5′-AGGTCGGTGTGAACG-GATTTG-3′ and reverse, 5′-GGGGTCGTTGATGGCAAC-3′.

## Ribo-seq data processing

The ribosome profiling and RNA sequencing data of immune cells (SRR8130804, SRR8130805, SRR7956051, SRR6189658) were obtained from the GEO database. For Ribo-Seq data analysis, adapter sequences were first trimmed with the FASTX-toolkit (http://hannonlab.cshl.edu/fastx_toolkit/), retaining only sequences that are at least 28 nt long. Trimmed Ribo-Seq reads aligning to tRNA and rRNA sequences were then removed using STAR v2.5.2b. Next, the remaining Ribo-Seq reads were aligned to the GRCm10 mouse genome assembly using STAR. We generated read alignments to the respective reference genomes for Ribo-seq libraries in BAM (binary version of sequence alignment map format). The genomic view of reads in Ribo-seq for Dleu2-17aa ORF is viewed in the IGV browser.

## Statistical analysis

The data were analyzed with GraphPad Prism 8 and are presented as the mean ± SEM. Student's t-test or two-way ANOVA was performed to determine statistical significance. $P < 0.05$ was considered statistically significant (n.s. indicates not significant, $*P < 0.05$, $**P < 0.01$, and $***P < 0.001$).

# Data availability

The mass spectrometry data have been deposited to the ProteomeX-change Consortium via the PRIDE partner repository with the dataset identifier PXD046468. The raw data of the flow cytometry screen are available in FlowRepository (http://flowrepository.org) with the Repository ID FR-FCM-Z7YC. http://flowrepository.org/id/RvFrGAqe1hNtmygpTcIDpafscddEary15MnysEJv2DG57fjsAPiSgR6-damOYwZIC.

# Peer review information

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

## Acknowledgements

This work was supported by the National Natural Science Foundation of China Original Exploration Program (No. 82050009), the National Key Research and Development Program of the Ministry of Science and Technology (2020YFA0112900), the National Science Fund of China (No. 81930088, 82203914, 82070509 [Y.S.], 82101909 [H.Z.], 82073428 [J.B.], and 82103719 [S.D.]), Clinical Research Plan of Shanghai Shenkang Hospital Development Center (SHDC2020CR3061B), SJTU Trans-med Awards Research (20210102), Integrated innovation fund of Shanghai Jiao Tong University (2021JCPT04), Experimental Animal Research Project of "Scientific and Technological Innovation Action Plan" (22140903100), Shanghai Action Plan for Science, Technology and Innovation (22QA1407600), Shanghai Action Plan for Science, Technology and Innovation (23ZR1480700) and Innovative Research Team of High-Level Local Universities in Shanghai [by Honglin W, if not otherwise noted]. The synopsis image was created using Biorender, adapted from "Smad7 Gene Delivery Prevents Muscle Wasting Associated with Cancer in Mice", by BioRender.com (2023). Retrieved from https://app.biorender.com/biorender-templates/figures/all.

## Author contributions

**Sibei Tang**: Conceptualization; Data curation; Formal analysis; Investigation; Methodology; Writing—original draft; Writing—review and editing. **Junxun Zhang**: Conceptualization; Data curation; Formal analysis; Investigation; Methodology; Writing—original draft; Writing—review and editing. **Fangzhou Lou**: Funding acquisition; Methodology; Writing—review and editing. **Hong Zhou**: Funding acquisition; Methodology; Writing—review and editing. **Xiaojie Cai**: Formal analysis; Investigation; Methodology; Writing—review and editing. **Zhikai Wang**: Formal analysis; Investigation; Methodology; Writing—review and editing. **Libo Sun**: Formal analysis; Investigation; Methodology; Writing—review and editing. **Yang Sun**: Funding acquisition; Methodology; Writing—review and editing. **Xiangxiao Li**: Formal analysis; Investigation; Methodology; Writing—review and editing. **Li Fan**: Formal analysis; Investigation; Writing—review and editing. **Yan Li**: Formal analysis; Investigation; Methodology; Writing—review and editing. **Xinping Jin**: Formal analysis; Investigation; Writing—review and editing. **Siyu Deng**: Funding acquisition; Methodology;

Writing—review and editing. **Qianqian Yin**: Formal analysis; Investigation; Methodology; Writing—review and editing. **Jing Bai**: Funding acquisition; Methodology; Writing—review and editing. **Hong Wang**: Methodology; Writing—review and editing. **Honglin Wang**: Conceptualization; Supervision; Funding acquisition; Validation; Project administration; Writing—review and editing.

## Disclosure and competing interests statement

The authors declare no competing interests.

# Expanded View Figures

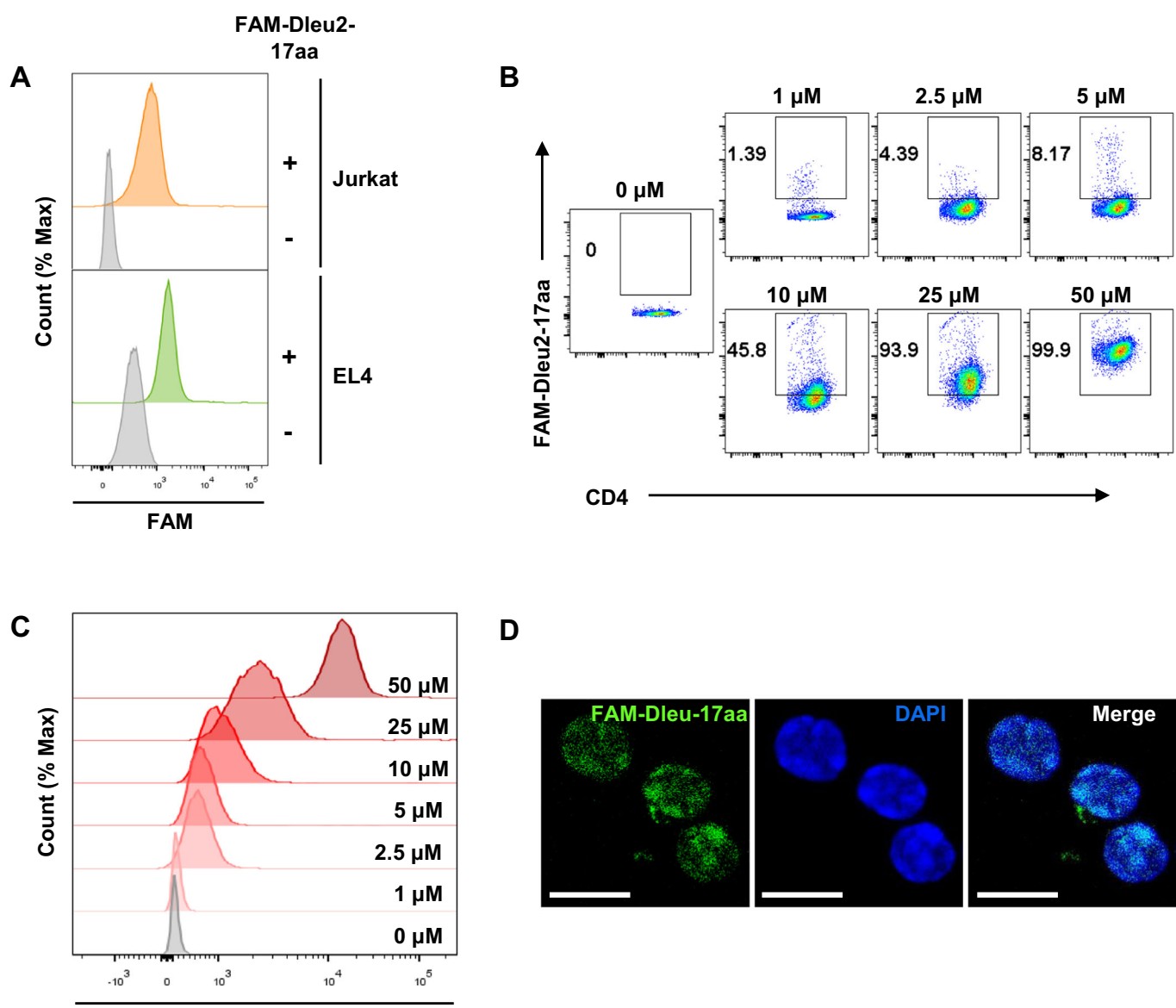

**Figure EV1.  Dleu2-17aa enters CD4+ T cells.**

(**A**) Representative histogram of Dleu2-17aa (10 μM) uptake in Jurkat and EL4 cell lines (n = 3). (**B**) Flow cytometry plots showing FAM-Dleu2-17aa+ cells gated on CD4+ T cells after treatment with increasing concentrations of FAM-Dleu2-17aa for 72 h (n = 3). (**C**) Histograms showing fluorescence intensity of FAM-Dleu2-17aa in CD4+ T cells after treatment with increasing concentrations of FAM-Dleu2-17aa for 72 h. Gray filled: untreated control (n = 3). (**D**) Immunofluorescence imaging of FAM-Dleu2-17aa (10 μM) penetration by CD4+ T cells isolated from WT mouse spleen (n = 3). Scale bars: 20 μm. Data information: n indicates biological replicate.

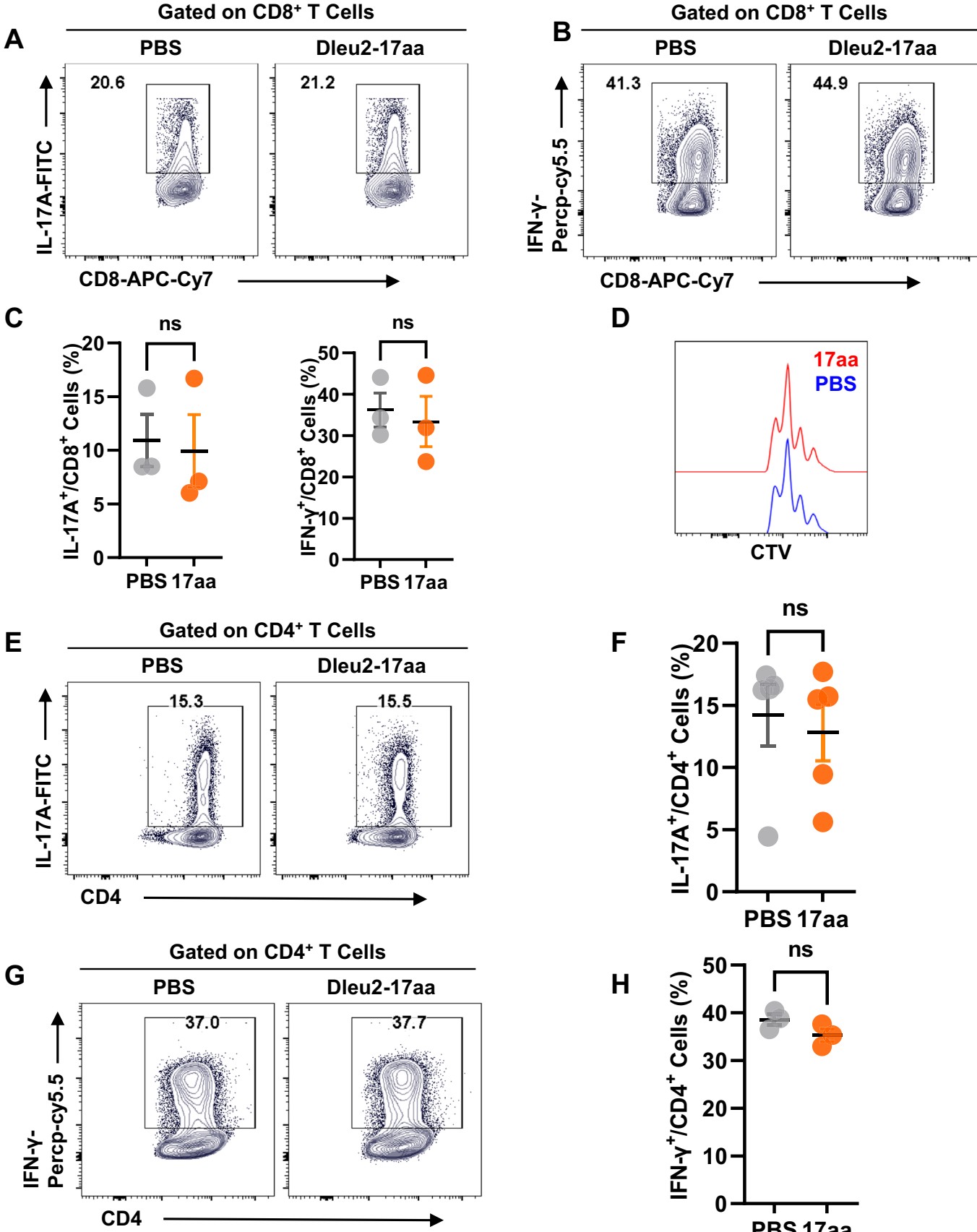

**Figure EV2. Effects of Dleu2-17aa on CD8$^+$ T cells and Th17 and Th1 cell differentiation.**

(A–C) Phenotype (A,B) and relative frequencies (C) of IL-17A$^+$ Tc17 cells (A), and IFN-γ$^+$ Tc1 cells (B) subsets in PBS and Dleu2-17aa treated CD8$^+$ T cells of WT mice. Sorted naive CD8$^+$ T cells were incubated with PBS or Dleu2-17aa (10 μM) for 72 h under Tc1 or Tc17 induction conditions before analyzing cytokine expression in live cells (n = 3). (D) Representative flow cytometry of CTV histograms for CD8$^+$ T cells stimulated with anti-CD3 and anti-CD28 and cultured with either PBS or Dleu2-17aa (10 μM) for 72 h (n = 3). (E–H) Phenotype (left panel) and relative frequencies (right panel) of IL-17A$^+$ Th17 cells (E,F), IFN-γ$^+$ Th1 cells (G,H) subsets in PBS and Dleu2-17aa treated CD4$^+$ T cells of WT mice. Sorted naive CD4$^+$ T cells were incubated with PBS or Dleu2-17aa (10 μM) for 72 h under Th1 or Th17 induction conditions before analyzing cytokine expression in live cells (n = 3–5). Data information: n indicates biological replicate. Error bars are mean ± SEM; n.s. indicates no significant difference. Statistical analysis was by two-tailed Student's t-test for (C,F,H).

**A**

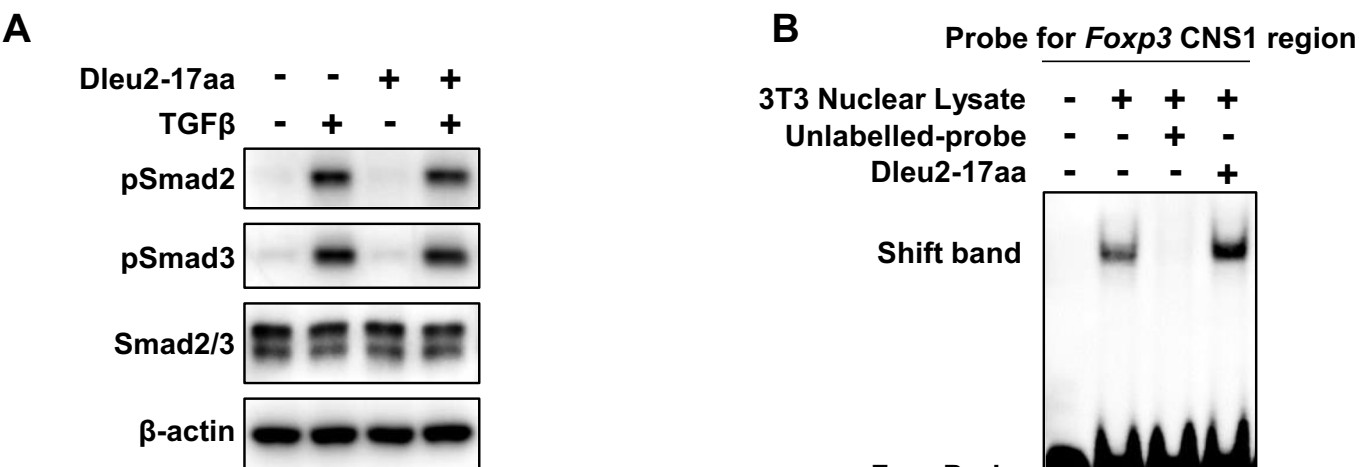

**B**

Probe for *Foxp3* CNS1 region

| | | | | |
|---|---|---|---|---|
| Dleu2-17aa | - | - | + | + |
| TGFβ | - | + | - | + |

pSmad2

pSmad3

Smad2/3

β-actin

| | | | | |
|---|---|---|---|---|
| 3T3 Nuclear Lysate | - | + | + | + |
| Unlabelled-probe | - | - | + | - |
| Dleu2-17aa | - | - | - | + |

Shift band

Free-Probe

**C**

2086
|

ATGAGATAACTGTTCAC**CCCATGTTGGCTTCCAGTCTCCTTTATGG**CTTCATTTTTTCCATTTAC
               **ChIP-qPCR-F**

2151
|
                                    **Smad3 binding region**

TGCAGAGGTCAAAAGTGTGGG**TATGGGAGCCAGACTGTCTGGAACAACCT**AGCCTCAACTCAAGT
                     **EMSA-Probe**

2216
|

CATCTGTGTGAATTTT**ACCCAGGCTCTTAACCTCTCTGTACCT**CCATTTCCTCGTATGTACTGTG
               **ChIP-qPCR-R**

**D**

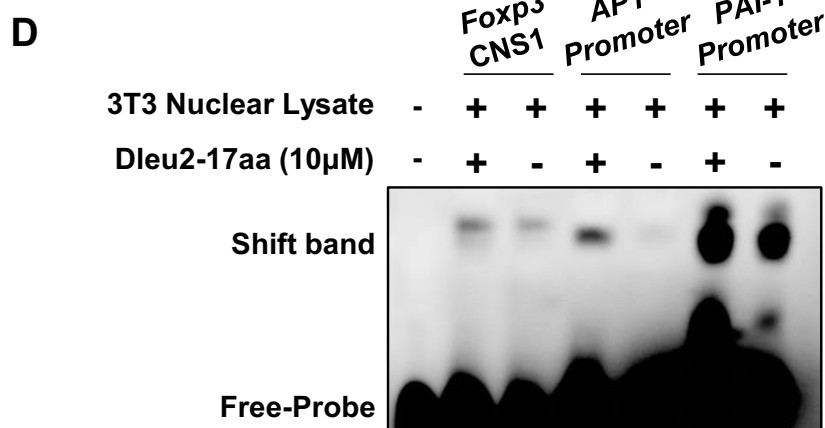

| | | Foxp3 CNS1 | | AP1 Promoter | | PAI-1 Promoter | |
|---|---|---|---|---|---|---|---|
| 3T3 Nuclear Lysate | - | + | + | + | + | + | + |
| Dleu2-17aa (10μM) | - | + | - | + | - | + | - |

Shift band

Free-Probe

Figure EV3.    Dleu2-17aa promotes Smad3 binding to the *Foxp3* CNS1 region.

(A) Western blot showing Total Smad2/Smad3 (lower blot) and phosphorylation of Smad2/Smad3 (upper blot) in CD4+ T cells treated with TNF-β and/or Dleu2-17aa ($n = 3$). (B) EMSA was performed in NIH/3T3 cell nuclear lysates using a 10 μM biotin-labeled Foxp3 probe and treated with Dleu2-17aa (10 μM) or vehicle control. There were band shifts to a higher molecular weight and an increase in band intensity after Dleu2-17aa treatment ($n = 3$). (C) The CNS1 region of the *Foxp3* gene where Smad3 binds to. The EMSA probe and ChIP-qPCR primers used were pointed out in the figure. The Smad3 binding region was highlighted in blue. (D) EMSA was performed to assess the binding interactions between Dleu2-17aa and specific DNA sequences within NIH/3T3 cell nuclear lysates. 10 μM of different biotin-labeled probe were employed for the assay. Following treatment with Dleu2-17aa (10 μM) or vehicle control. There were band shifts to a higher molecular weight and an increase in band intensity after Dleu2-17aa treatment ($n = 3$). Data information: n indicates biological replicate.

**A**

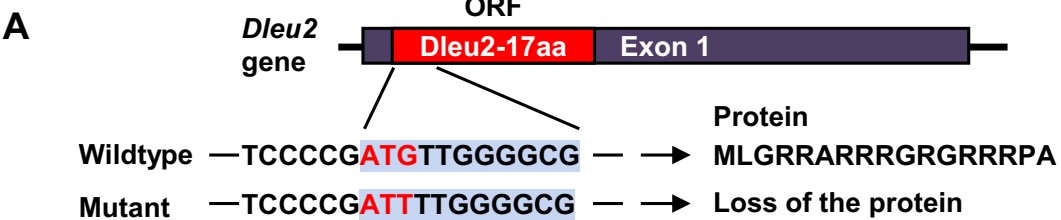

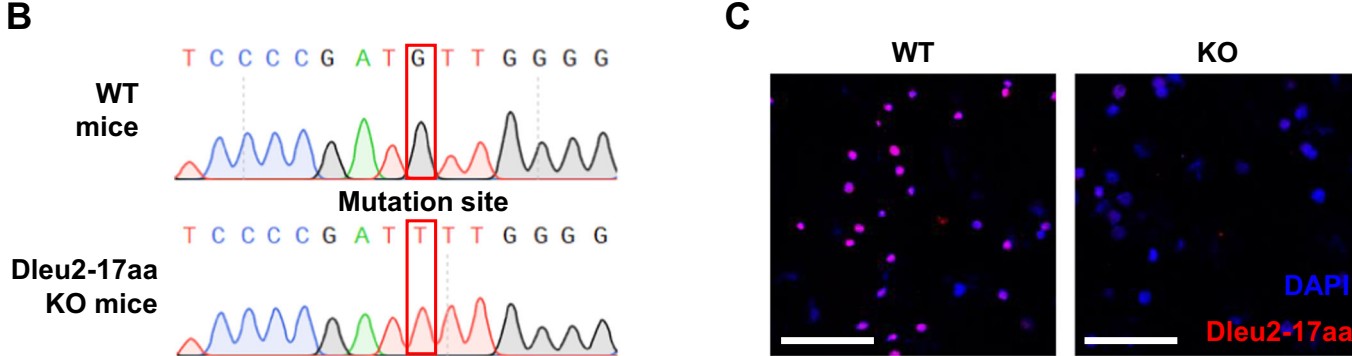

**Figure EV4.  Construction of Dleu2-17aa knockout mice.**

(**A**) Schematic of the Dleu2-17aa mutant construct. The start codon ATG of Dleu2-17aa was mutated into ATT. (**B**) Sanger sequencing result of Dleu2-17aa locus from genomic DNA of the WT and KO mouse. (**C**) Representative immunofluorescence imaging of endogenous Dleu2-17aa expression in CD4[+] T cells of WT and KO mice (*n* = 3). Scale bars: 100 μm. Data information: n indicates biological replicate.

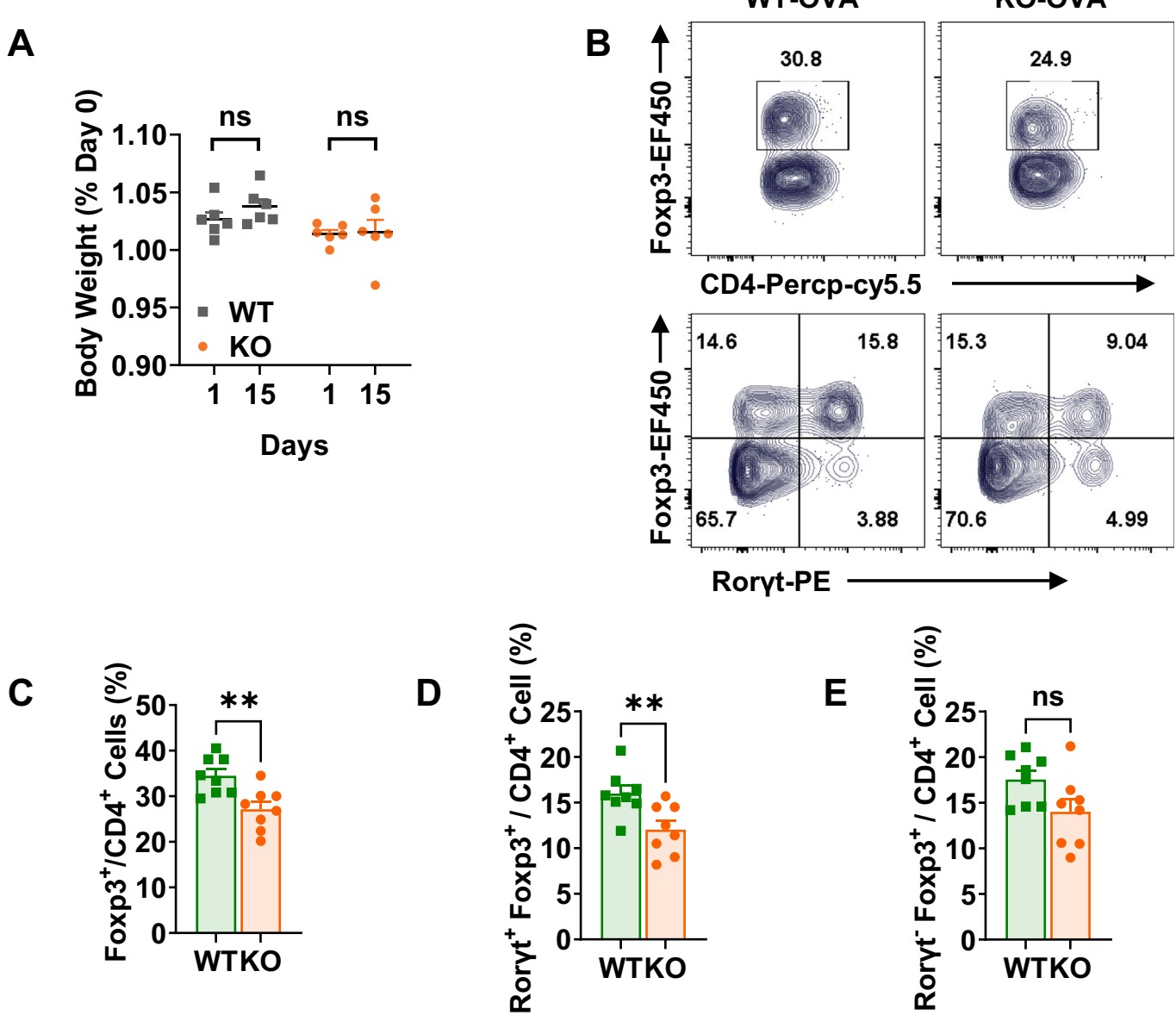

**Figure EV5. Effects of Dleu2-17aa on dietary antigen induced pTreg differentiation.**

(A) Body weight of day 1 and day 15 after oral gavage with OVA antigen and CTB ($n = 6$–8). (B) Representative flow plots showing Foxp3 and RORγt expression gated on CD4$^+$ cells from colon of WT and KO mice after 15 days OVA and CTB oral gavage ($n = 6$–8). (C–E) Summary graph depicting the percentage of Foxp3$^+$ cells (C), RORγt$^+$Foxp3$^+$ cells (D), RORγt$^-$Foxp3$^+$ cells (E) among total CD4$^+$ T cells in WT and KO mice cLN after 15 days OVA and CTB oral gavage ($n = 6$–8). Data information: n indicates biological replicate. Error bars are mean ± SEM; **$P < 0.01$, n.s. indicates no significant difference. Statistical analysis was by two-tailed Student's t-test for (A,C,D,E).

