## [Peer Review File · EMBO Reports]

A lncRNA Dleu2-encoded peptide relieves autoimmunity by facilitating Smad3-mediated Treg induction

Honglin Wang, Sibeï TANG, Junxun ZHANG, Fangzhou Lou, Hong Zhou, Xiaojie CAI, Zhikai Wang, Libo SUN, Yang Sun, Xiangxiao LI, Li FAN, Yan Li, Xinping Jin, Siyu Deng, Qianqian YIN, Jing Bai, and Hong Wang

Corresponding author(s): Honglin Wang (honglin.wang@sjtu.edu.cn)

Review Timeline:

Submission Date:	15th May 23
Editorial Decision:	5th Jul 23
Revision Received:	4th Nov 23
Editorial Decision:	28th Nov 23
Revision Received:	2nd Jan 24
Accepted:	9th Jan 24

Editor: Achim Breiling

Transaction Report:

Dear Prof. Wang,

Thank you for the transfer of your manuscript to EMBO reports. I have now received the reports from the three referees that were asked to evaluate your study, which can be found at the end of this message.

As you will see, the referees state that these findings are of high interest. However, they all have comments and suggestions to improve the study, indicating that a major revision of the manuscript is necessary to allow publication of the study in EMBO reports. As the reports are below, and all the referee concerns need to be addressed, I will not detail them here.

Given the constructive referee comments, I would like to invite you to revise your manuscript with the understanding that all referee concerns must be addressed in the revised manuscript and in a detailed point-by-point response. Acceptance of your manuscript will depend on a positive outcome of a second round of review. It is EMBO reports policy to allow a single round of revision only and acceptance of the manuscript will therefore depend on the completeness of your responses included in the next, final version of the manuscript.

- 1) a .docx formatted version of the final manuscript text (including legends for main figures, EV figures and tables), but without the figures included. Figure legends should be compiled at the end of the manuscript text.
- 2) individual production quality figure files as .eps, .tif, .jpg (one file per figure), of main figures (up to 8) and EV figures (up to 5). Please upload these as separate, individual files upon re-submission.

- 4) a complete author checklist, which you can download from our author guidelines (<https://www.embopress.org/page/journal/14693178/authorguide>). Please insert page numbers in the checklist to indicate where the requested information can be found in the manuscript. The completed author checklist will also be part of the RPF.

- 5) that primary datasets produced in this study (e.g. RNA-seq, ChIP-seq, structural and array data) are deposited in an

appropriate public database. If no primary datasets have been deposited, please also state this in a dedicated section (e.g. 'No primary datasets have been generated and deposited'), see below.

The accession numbers and database should be listed in a formal "Data Availability" section (placed after Materials & Methods) that follows the model below. This is now mandatory (like the COI statement). Please note that the Data Availability Section is restricted to new primary data that are part of this study. This section is mandatory. As indicated above, if no primary datasets have been deposited, please state this in this section

Data availability

8) Regarding data quantification and statistics, please make sure that the number "n" for how many independent experiments were performed, their nature (biological versus technical replicates), the bars and error bars (e.g. SEM, SD) and the test used to calculate p-values is indicated in the respective figure legends (also for potential EV figures and all those in the final Appendix). Please also check that all the p-values are explained in the legend, and that these fit to those shown in the figure. Please provide statistical testing where applicable. Please avoid the phrase 'independent experiment', but clearly state if these were biological or technical replicates. Please also indicate (e.g. with n.s.) if testing was performed, but the differences are not significant. In case n=2, please show the data as separate datapoints without error bars and statistics. See also: <http://www.embopress.org/page/journal/14693178/authorguide#statisticalanalysis>

9) Please add scale bars of similar style and thickness to microscopic images, using clearly visible black or white bars (depending on the background). Please place these in the lower right corner of the images themselves. Please do not write on or near the bars in the image but define the size in the respective figure legend.

10) Please also note our reference format:

12) We now use CRedit to specify the contributions of each author in the journal submission system. CRedit replaces the author contribution section. Please use the free text box to provide more detailed descriptions and do not provide your final manuscript text file with an author contributions section. See also our guide to authors: <https://www.embopress.org/page/journal/14693178/authorguide#authorshipguidelines>

Finally, please order the manuscript sections like this, using these names:

Title page - Abstract - Keywords - Introduction - Results - Discussion - Materials and Methods - Data availability section - Acknowledgements - Disclosure and Competing Interests Statement - References - Figure legends - Expanded View Figure legends

I look forward to seeing a revised version of your manuscript when it is ready. Please let me know if you have questions or comments regarding the revision.

Yours sincerely,

Referee #1:

The authors tried to show that lncRNA Dleu2-Encoding Peptide attenuates EAE by facilitating Smad3-mediated Treg induction. They showed that lncRNA Dleu2 encoded Dleu2-17aa, which promoted inducible regulatory T (iTreg) cell generation through interacting with SMAD Family Member 3 (Smad3) and enhancing the binding of Smad3 to the Foxp3 conserved non-coding DNA sequence 135 (CNS1) region. Moreover, they showed that genetic deletion of Dleu2-17aa in mice through start codon mutation impaired iTreg generation and exacerbated EAE, and that exogenously supplemented Dleu2-17aa attenuated EAE. The manuscript is well written.

The topic is important. However, the results of EAE experiments are premature and not convincing. There are several major concerns.

1. Figure 5 B shows extremely severe demyelination in scrPEP-treated control EAE mice. However, Figure 7C shows mild demyelination in WT control EAE mice, and moderate demyelination in Dleu2-17aa KO mice with EAE. In fact, there is much more severe demyelination in scrPEP-treated control EAE mice than Dleu2-17aa KO mice with EAE. How to explain this very confusing result?
2. The authors should perform experiments (such as in vitro recall assay) to determine whether Dleu2-17aa KO or exogenously supplemented Dleu2-17aa affect T cell priming and reactivation in the EAE model.
3. The authors used global Dleu2-17aa KO or exogenously supplemented Dleu2-17aa. Does Dleu2-17aa express in neurons and glia (oligodendrocytes, neurons, microglia, and astrocytes)? All these cell types play critical roles in the pathogenesis of EAE. Does global Dleu2-17aa KO or exogenously supplemented Dleu2-17aa affect the activity of microglia, and astrocytes during EAE? Does global Dleu2-17aa KO or exogenously supplemented Dleu2-17aa affect the viability of neurons or oligodendrocytes?
4. The authors should quantify inflammation, demyelination, oligodendrocyte loss, and axon degeneration in the CNS of EAE models.

Referee #2:

The work entitled « lncRNA Dleu2-Encoding Peptide Relieves Autoimmunity by Facilitating Smad3-mediated Treg Induction » by Sabei TANG and collaborators identifies a small peptide produced from a short open reading frame located in the lncRNA Dleu2 transcript. They described its expression and function.

- They first provide experimental evidences that Dleu2 encodes a 17-amino-acid micropeptide.
- The micropeptide is abundantly expressed in T cells
- It promoted inducible regulatory T (iTreg) cell generation in vitro
- Mechanistically, they show that Dleu2 micropeptide interact with SMAD Family Member 3 (Smad3) and this results in an increase of the binding of Smad3 to a promoter region of the Foxp3 gene.
- This consequently result in a better activation of the Foxp3 target genes upon TGFβ activation.
- They performed start codon mutation to generate KO of this peptide in mice. Its mutation compromised iTreg generation and exacerbated experimental autoimmune encephalomyelitis (EAE).

The study is interesting and is in the cope of EMBO reports. I am not specialist of Treg cells biology but the experiments provided look convincing, their conclusions correctly supported. The Dleu2 peptide characterization is well done and the data are convincing.

I have therefore no special concerns regarding the manuscript publication.

Other points:

- English grammar (tenses and conjugations) should be checked
- MS data must be made available. As well, an excel file describing each protein identified must be provided.
- I have also one suggestion to strengthen the story. I am wondering whether the Dleu2 peptide enhance Smad binding only on the Foxp3 DNA region. Is this specific of this sequence or is the Smad binding increased on all binding sites targeted? Are some adjacent nucleotides involved? Using a synthetic Smad3 consensus-binding site (or binding from other promoters/enhancers) and testing if addition of the Dleu2 peptide also enhances the Smad3 binding can test this.
- lane 160. The other interactors should be described and listed (how many, names etc)
- lane 222. Typo; "its" but not "tis"

Referee #3:

The authors identified a novel 17aa short peptide encoded by Dleu2, a previously designated lncRNA, that is important for Smad3 binding to the CNS1 enhancer of Foxp3 locus. Mice treated with Dleu2 induced more Tregs during EAE and had better outcome, while deletion of Dleu2 resulted in impaired Treg induction and worse clinical scores. The study is elegantly designed and well performed with a novel discovery, however, there are certain issues with how data was presented and the quality of the figures, and here are the specifics:

1. In Figure 2D, the representative plots are all selected from the samples with the lowest percentage. Especially for 5uM and 10uM groups, the "representative plots" can even be considered outliers.
 - a. While treating cells with the small peptide has therapeutic implications, dose dependent uptake needs to be demonstrated.
 - b. Maybe over expressing Dleu2 in cells and compare low expressers and high expressers for their Foxp3 induction potential could be a better experiment to perform.
2. In Figure 4B, what cis element is used and in what cell type? If the main focus of this manuscript is on CNS1, should a CNS1 reporter be used in this experiment?
3. In Figure 6G, Treg population in barrier tissues should be shown and their expression of RORgt analyzed. pTregs are especially abundant in the gut, where RORgt expression marks the microbiota induced pTregs. CNS1 deficient mice have been shown to harbor fewer Tregs in the intestine.
4. In addition to EAE, it would also be helpful to show if there is a difference between WT and KO animals in pTreg induction under homeostatic conditions, i.e in response to dietary antigen with OVA feeding etc.
5. Are Figures 5E and 7E done in a similar fashion? The cytokine production profile (IFN γ production and IL-17 production) looks drastically different between the two experiments. It might affect how the data is interpreted.

POINT-TO-POINT RESPONSE TO THE REVIEWERS' COMMENTS

Changes in the revised manuscript text are underlined.

Reviewers' Comments:

Referee #1:

The authors tried to show that lncRNA Dleu2-Encoding Peptide attenuates EAE by facilitating Smad3-mediated Treg induction. They showed that lncRNA Dleu2 encoded Dleu2-17aa, which promoted inducible regulatory T (iTreg) cell generation through interacting with SMAD Family Member 3 (Smad3) and enhancing the binding of Smad3 to the Foxp3 conserved non-coding DNA sequence 1 35 (CNS1) region. Moreover, they showed that genetic deletion of Dleu2-17aa in mice through start codon mutation impaired iTreg generation and exacerbated EAE, and that exogenously supplemented Dleu2-17aa attenuated EAE. The manuscript is well written.

The topic is important. However, the results of EAE experiments are premature and not convincing. There are several major concerns.

Comment 1. Figure 5 B shows extremely severe demyelination in scrPEP-treated control EAE mice. However, Figure 7C shows mild demyelination in WT control EAE mice, and moderate demyelination in Dleu2-17aa KO mice with EAE. In fact, there is much more severe demyelination in scrPEP-treated control EAE mice than Dleu2-17aa KO mice with EAE. How to explain this very confusing result?

Re: We are deeply grateful to the reviewer for catching this inappropriate image selection. We thoroughly re-examined the original images and provided pictures consistent with the statistics of demyelination. (**revised Fig. 5C, 7C**) Here we present more original LFB staining images of these experiments.

Comment 2. The authors should perform experiments (such as *in vitro* recall assay) to determine whether Dleu2-17aa KO or exogenously supplemented Dleu2-17aa affect T cell priming and reactivation in the EAE model.

Re: We sincerely appreciate the valuable suggestion provided by the reviewer. Following the reviewer's suggestion, we performed *in vitro* T cell recall assay. Specifically, splenic CD4⁺ T cells were isolated from WT and KO mice following EAE induction. Then the CD4⁺ T cells were re-stimulated with the myelin oligodendrocyte glycoprotein (MOG)₃₃₋₃₅ peptide in the presence of irradiated antigen presenting cells from Rag1^{-/-} mice spleen. Using this approach, we measured cytokine production by intracellular staining and flow cytometric analysis. No differences were observed in the frequencies of IL-17A⁺ or IFN- γ ⁺ CD4⁺ T cells between WT and KO groups (**revised Appendix Fig. S7A**). Proliferative capacity was also assessed and found to be comparable between two groups upon MOG rechallenge, as demonstrated by cell trace violet dilution assays (**revised Appendix Fig. S7B**). These implied Dleu2-17aa modulates EAE pathogenesis by regulating Treg induction rather than T cell reactivation.

Comment 3. The authors used global Dleu2-17aa KO or exogenously supplemented Dleu2-17aa. Does Dleu2-17aa express in neurons and glia (oligodendrocytes, neurons, microglia, and astrocytes)? All these cell types play critical roles in the pathogenesis of EAE. Does global Dleu2-17aa KO or exogenously supplemented Dleu2-17aa affects the activity of microglia, and astrocytes during EAE? Does global Dleu2-17aa KO or exogenously supplemented Dleu2-17aa affects the viability of neurons or oligodendrocytes?

Re: We would like to express our sincere gratitude to the reviewer's thoughtful suggestion to examine Dleu2-17aa localization within the CNS. Following the reviewer's guidance, we conducted immunofluorescent co-staining experiments to evaluate colocalization of Dleu2-17aa

with various cell types, including CD4⁺ T cells, microglia, astrocytes, and neurons in spinal cord sections from Dleu2-17aa and scrambled peptide-treated EAE mice. The results revealed Dleu2-17aa is expressed in almost all the CD4⁺ T cell. Furthermore, we observed some colocalization with microglia, limited colocalization with astrocytes, and minimal colocalization with neurons. While supplementation of Dleu2-17aa conferred some protective effects, it did not substantially alter activity or viability of glia or neurons, as the morphology did not change (**revised Appendix Fig. S6**). However, we did observe reduced CD4⁺ T cell infiltration in the spinal cords of Dleu2-17aa-treated EAE mice compared with scrPEP controls. We extend our heartfelt thanks to the reviewer for the valuable time and expertise in reviewing our manuscript and for this analysis that broadens CNS characterization.

Comment 4. The authors should quantify inflammation, demyelination, oligodendrocyte loss, and axon degeneration in the CNS of EAE models.

Re: We appreciate the reviewer's insightful suggestion. As advised, we quantified inflammatory foci and demyelinated lesions in the CNS using scoring systems, which are commonly used and well-established parameters in EAE research (**revised Figures 7D and 5D**). The data revealed increased pathology in KO and WT and Dleu2-17aa supplemented EAE mice (**revised Figures 7D and 5D**). We are grateful to the reviewer for recommending these important analyses.

Referee #2:

The work entitled « LncRNA Dleu2-Encoding Peptide Relieves Autoimmunity by Facilitating Smad3-mediated Treg Induction » by Sibeï TANG and collaborators identifies a small peptide produced from a short open reading frame located in the LncRNA Dleu2 transcript. They described its expression and function.

- They first provide experimental evidences that Dleu2 encodes a 17-amino-acid micropeptide.
- The micropeptide is abundantly expressed in T cells
- It promoted inducible regulatory T (iTreg) cell generation in vitro
- Mechanistically, they show that Dleu2 micropeptide interact with SMAD Family Member 3 (Smad3) and this results in an increase of the binding of Smad3 to a promoter region of the Foxp3 gene.
- This consequently result in a better activation of the Foxp3 target genes upon TGFβ activation.
- They performed start codon mutation to generate KO of this peptide in mice. Its mutation compromised iTreg generation and exacerbated experimental autoimmune encephalomyelitis (EAE).

The study is interesting and is in the cope of EMBO reports. I am not specialist of Treg cells biology but the experiments provided look convincing, their conclusions correctly supported. The

Dleu2 peptide characterization is well done and the data are convincing.

I have therefore no special concerns regarding the manuscript publication.

Other points:

Comment - English grammar (tenses and conjugations) should be checked

Re: We would like to sincerely thank the reviewer for the valuable feedback to improve the clarity and precision of our manuscript's language. As suggested, we have thoroughly re-examined the English writing to verify accurate grammar, consistent tense usage, and correct verb conjugation throughout the text.

Comment - MS data must be made available. As well, an excel file describing each protein identified must be provided.

Re: We would like to thank the reviewer for prompting us to better disclose how readers can access the raw data underlying this work. The mass spectrometry raw data files have been deposited to the ProteomeXchange Consortium via the PRIDE partner repository with the dataset identifier **PXD046468**. In addition, **Appendix Table S1** provides full details of all proteins identified in the proteomic analysis, including accession numbers, peptides detected, and protein quantification values.

Comment - I have also one suggestion to strengthen the story. I am wondering whether the Dleu2 peptide enhance Smad binding only on the Foxp3 DNA region. Is this specific of this sequence or is the Smad binding increased on all binding sites targeted? Are some adjacent nucleotides involved? Using a synthetic Smad3 consensus-binding site (or binding from other promoters/enhancers) and testing if addition of the Dleu2 peptide also enhances the Smad3 binding can test this.

Re: We appreciate the reviewer for raising an important point regarding the specificity of Dleu2-17aa's effect on Smad3 binding activity. To directly address this, we conducted electrophoretic mobility shift assays (EMSAs) using synthetic oligonucleotide probes containing three distinct Smad3 binding site consensus sequences: the *Foxp3* CNS1 region, *AP-1* promoter, and *PAI-1* promoter. We found that Smad3 binding was enhanced similarly for all three probes tested upon Dleu2-17aa addition (**revised Fig. EV3D**), this suggested that Dleu2-17aa promoted Smad3 activity as a transcriptional factor.

Comment - lane 160. The other interactors should be described and listed (how many, names etc)

Re: We appreciate the reviewer's feedback to make the dataset completed and more useful to other researchers. In line with the reviewer's suggestion, we provided a comprehensive list of the other identified interactors as supplementary data (**revised Appendix Table S1**). Our LC-MS analysis revealed a total of 431 candidate interacting proteins.

Comment - lane 222. Typo; "its" but not "tis"

Re: We deeply appreciate the reviewer for the careful review of our manuscript. The typo on line 222 noted by the reviewer has been corrected.

Referee #3:

The authors identified a novel 17aa short peptide encoded by Dleu2, a previously designated lncRNA, that is important for Smad3 binding to the CNS1 enhancer of Foxp3 locus. Mice treated with Dleu2 induced more Tregs during EAE and had better outcome, while deletion of Dleu2 resulted in impaired Treg induction and worse clinical scores. The study is elegantly designed and well performed with a novel discovery, however, there are certain issues with how data was presented and the quality of the figures, and here are the specifics:

Comment 1. In Figure 2D, the representative plots are all selected from the samples with the lowest percentage. Especially for 5uM and 10uM groups, the "representative plots" can even be considered outliers.

- a. While treating cells with the small peptide has therapeutic implications, dose dependent uptake needs to be demonstrated.
- b. Maybe over expressing Dleu2 in cells and compare low expressers and high expressers for their Foxp3 induction potential could be a better experiment to perform.

Re: We deeply appreciate the reviewer for the careful review of our manuscript and for noting the plots in Figure 2D were not fully representative. We have updated the figure with more typical examples from each dose group.

We would like to sincerely thank the reviewer for raising these important issues as demonstrating dose-dependent uptake of the Dleu2-17aa helps strengthen our conclusions. In the revised manuscript, we showed dose-dependent penetration of fluorescence labeled Dleu2-17aa by CD4⁺ T cells, with maximal uptake observed between 25-50 μ M (**revised Fig. EV1B and C**).

Additionally, as suggested, we have performed new experiments overexpressing Dleu2-17aa in T cells using a lentiviral vector. We sorted cells into medium and high Dleu2-17aa-expressing populations and induced iTreg differentiation. We found that high Dleu2-17aa-expressing T cells showed increased iTreg differentiation compared with low expressing T cells (**revised Appendix Fig S3**).

Comment 2. In Figure 4B, what cis element is used and in what cell type? If the main focus of this manuscript is on CNS1, should a CNS1 reporter be used in this experiment?

Re: We are extremely grateful to the reviewer for pointing out our oversight. In the revised manuscript, we have clarified the details of the luciferase reporter assays evaluating the effect of

Dleu2-17aa on Smad3-mediated transcription. Specifically, we constructed a luciferase reporter plasmid containing the *Foxp3* CNS1 region and transfected this reporter into NIH 3T3 cells along with a Smad3 expression vector. We observed that treatment with Dleu2-17aa led to a significant increase in CNS1-dependent luciferase expression compared with control. This indicated that Dleu2-17aa can enhance Smad3 transcription ability on the CNS1 element.

Comment 3. In Figure 6G, Treg population in barrier tissues should be shown and their expression of ROR γ t analyzed. pTregs are especially abundant in the gut, where ROR γ t expression marks the microbiota induced pTregs. CNS1 deficient mice have been shown to harbor fewer Tregs in the intestine.

Re: We are grateful to the reviewer for the constructive feedback to help us enhance our work. Following the reviewer's valuable suggestion, we examined the pTreg population and ROR γ t expression in the colon lamina propria of WT and Dleu2-17aa KO mice, as this tissue harbors abundant ROR γ t⁺ pTregs. Using flow cytometry, we quantified Foxp3⁺ROR γ t⁺ double positive pTregs in the colonic lamina propria lymphocytes from WT and KO mice. We detected comparable frequencies of ROR γ t⁺ pTregs between the two group (**revised Figure 6I, J**), indicating that Dleu2-17aa deficiency does not alter ROR γ t⁺ pTreg homeostasis in the colon under steady state.

Comment 4. In addition to EAE, it would also be helpful to show if there is a difference between WT and KO animals in pTreg induction under homeostatic conditions, i.e in response to dietary antigen with OVA feeding etc.

Re: We appreciate the reviewer for taking the time to provide such thoughtful comments and suggestions. Following the reviewer's kind advise, we examined pTreg induction of WT and KO mice under homeostatic conditions by using an oral ovalbumin (OVA) feeding model. We fed WT and KO mice with OVA antigen plus cholera toxin subunit B (CTB) as adjuvant. No weight loss was observed in OVA and CTB fed mice (**revised Fig. EV5A**). We then analyzed Foxp3⁺ROR γ t⁺ pTregs in the colonic lamina propria lymphocytes. With flow cytometry (**revised Fig. EV5B**) and quantification (**revised Fig. EV5C-E**), we observed that Dleu2-17aa KO mice showed a significant reduction in the frequency of pTregs compared with WT mice. This suggested that Dleu2-17aa promotes pTreg differentiation driven by dietary antigens. The impaired oral antigen-induced pTreg formation aligns with the exacerbated EAE phenotypes observed in Dleu2-17aa KO mice. Taken together, these data revealed an important function for Dleu2-17aa in pTreg generation upon dietary antigen stimulation, complementing its effect on iTregs driven by TGF- β signaling. We thank the reviewer for this invaluable suggestion to assess pTregs under physiological conditions, which uncovered a novel role for Dleu2-17aa in oral tolerance and mucosal pTreg responses.

Comment 5. Are Figures 5E and 7E done in a similar fashion? The cytokine production profile (IFN γ production and IL-17 production) looks drastically different between the two experiments. It might affect how the data is interpreted.

Re: We sincerely apologize for the confusion created by incorrect flow antibody label in Figures 5E. After the reviewer pointed this out, we realized the antibody markers in the flow plots were mistakenly swapped in Figure 5E. We have checked and replaced the Figure 5E with correct cytokine antibody label in the revised manuscript. We are extremely grateful to the reviewer for catching this issue and helping us re-evaluating our data to generate reliable data quality that will substantially strengthen the manuscript. Here we present original flow cytometry data in figure5E.

Dear Prof. Wang,

Thank you for the submission of your revised manuscript to our editorial offices. I have now received the reports from the three referees that I asked to re-evaluate your study, you will find below. As you will see, the referees now support the publication of the study in EMBO reports. Referees #1 and #3 have some remaining concerns or suggestions to improve the manuscript I ask you to address in a final revised manuscript.

- I would suggest this slightly modified title:

An lncRNA Dleu2-encoded peptide relieves autoimmunity by facilitating Smad3-mediated Treg induction

- Please provide the abstract written in present tense throughout.

- Please move the keywords below the abstract. Please add two more to have 5 final keywords.

- Please make sure that the number "n" for how many independent experiments were performed, their nature (biological versus technical replicates), the bars and error bars (e.g. SEM, SD) and the test used to calculate p-values is indicated in the respective figure legends (for main, EV and Appendix figures) of the final revised manuscript. Please also check that all the p-values are explained in the legend, and that these fit to those shown in the figure. Please provide statistical testing where applicable. Please avoid the phrase 'independent experiment', but clearly state if these were biological or technical replicates. Please also indicate (e.g. with n.s.) if testing was performed, but the differences are not significant. In case n=2, please show the data as separate datapoints without error bars and statistics. See also:

<http://www.embopress.org/page/journal/14693178/authorguide#statisticalanalysis>

If n<5, please show single datapoints for diagrams. It seems n.s. is missing from many diagrams. Moreover:

- Please note that in figures 4a there is a mismatch between the annotated p values in the figure legend and the annotated p values in the figure file that should be corrected.

- Please define the annotated p values ***/** in the legends of figures 4b-c as appropriate.

- Please indicate the statistical test used for data analysis in the legends of figures 4b-c; 6c.

- Please note that the error bars are not defined in the legends of figures 2a; 4b-c; 6c-d, h, j; 7b.

- Please add to each legend a 'Data Information' section explaining the statistics used or providing information regarding replicates and scales. See:

- We now use CRediT to specify the contributions of each author in the journal submission system. CRediT replaces the author contribution section. Please use the free text box to provide more detailed descriptions and do not provide your final manuscript text file with an author contributions section. See also our guide to authors:

<https://www.embopress.org/page/journal/14693178/authorguide#authorshippinguidelines>

- We updated our journal's competing interests policy in January 2022 and request authors to consider both actual and perceived competing interests. Please review the policy <https://www.embopress.org/competing-interests> and update your competing interests if necessary. Please name this section 'Disclosure and Competing Interests Statement' and put it after the Acknowledgements section.

- Please remove the sections "Corresponding author", the description of the synopsis image ("Synopsis: Graphical abstract of the main findings in this study" - BioRender can be acknowledged in the acknowledgements section) and the "Ethics declarations" from the manuscript text file. Or, if there are ethic declarations, please add these.

- Appendix Table S1 is a datasets. Please upload this as single excel file named Dataset EV1. Please add a title and a legend on the first TAB this file and please change the callouts of this item in the manuscript text file (Dataset EV1).

- Please make sure that all the funding information is also entered into the online submission system and that it is complete and similar to the one in the acknowledgement section of the manuscript text file. Presently, the Innovative Research Team of High-Level Local Universities in Shanghai is only mentioned in the acknowledgements.

- The Data Availability section should only contain information on large datasets that have been deposited to external repositories and all access information. Please remove any additional information from this section and make sure the datasets are public latest upon publication of the manuscript.

- Thanks for providing a schematic summary figure (synopsis image). Please provide this with bigger fonts and less blurry (but with the exact width of 550 pixels and a height of not more than 400 pixels).

Best,

Referee #1:

The authors are very responsive and address many concerns. The manuscript is significantly improved. However, there is a concern: The images of HE staining in Figure 5C and 7C are very hard to appreciate.

Referee #2:

the authors respond adequately to my requests.

Referee #3:

The authors have sufficiently addressed my concerns in this revision. One minor revision to further improve the quality of the manuscript is probably show flow cytometry analysis of the markers in Figure 4A as well.

point-to-point response addressing the editorial requests and the reviewers' comments

Changes in the revised manuscript text are highlighted.

Editorial requests:

- I would suggest this slightly modified title:

An lncRNA Dleu2-encoded peptide relieves autoimmunity by facilitating Smad3-mediated Treg induction

Re: We appreciate the suggested modification, and the revised title now reads as follows: "A lncRNA Dleu2-encoded peptide relieves autoimmunity by facilitating Smad3-mediated Treg induction."

- Please provide the abstract written in present tense throughout.

Re: We express our heartfelt gratitude for your valuable editorial requests, which have greatly contributed to the improvement of our manuscripts. We have revised the abstract to be consistently written in the present tense throughout the manuscript.

- Please move the keywords below the abstract. Please add two more to have 5 final keywords.

Re: We are truly grateful to for your help for your constructive feedback and suggestion. We have moved the keywords below the abstract as requested. Additionally, we have added two more keywords to ensure a total of five final keywords.

- Please make sure that the number "n" for how many independent experiments were performed, their nature (biological versus technical replicates), the bars and error bars (e.g. SEM, SD) and the test used to calculate p-values is indicated in the respective figure legends (for main, EV and Appendix figures) of the final revised manuscript. Please also check that all the p-values are explained in the legend, and that these fit to those shown in the figure. Please provide statistical testing where applicable. Please avoid the phrase 'independent experiment', but clearly state if these were biological or technical replicates. Please also indicate (e.g. with n.s.) if testing was performed, but the differences are not significant. In case n=2, please show the data as separate datapoints without error bars and statistics. See also:

<http://www.embopress.org/page/journal/14693178/authorguide#statisticalanalysis>

If $n < 5$, please show single datapoints for diagrams. It seems n.s. is missing from many diagrams. Moreover:

Re: We would like to express our deepest appreciation for meticulously reviewing our work and pointing out the mistakes and areas for improvement. We have thoroughly addressed the experimental details as your instructions. In the revised figure legends for main figures, EV figures, and Appendix figures, we have included the clarification of n as biological replicates, the error bar in every figure, the test used to calculate p-value, these information fit to those shown in the figure, we show data points in diagrams and added previously missing n.s..

- Please note that in figures 4a there is a mismatch between the annotated p values in the figure legend and the annotated p values in the figure file that should be corrected.

Re: We are deeply grateful for catching this inappropriate mismatch. We have carefully reviewed figure 4a and have corrected the mismatch between the annotated p-values in the figure legend and the annotated p-values in the figure file.

- Please define the annotated p values $^{***}/^{**}$ in the legends of figures 4b-c as appropriate.

Re: We are thankful for your careful attention to detail, which has helped us refine our research and present it more effectively. In the legends of figures 4b-c, we have appropriately defined the annotated p-values according to their significance.

- Please indicate the statistical test used for data analysis in the legends of figures 4b-c; 6c.

Re: We are genuinely appreciative of your dedication We have indicated the statistical tests used for data analysis in the legends of figures 4b-c and 6c.

- Please note that the error bars are not defined in the legends of figures 2a; 4b-c; 6c-d, h, j; 7b.

Re: We apologize for the oversight. In the revised legends of figures 2a, 4b-c, 6c-d, 6h, 6j, and 7b, we have now defined the error bars.

- Please add to each legend a 'Data Information' section explaining the statistics used or providing information regarding replicates and scales. See:

Re: We sincerely appreciate your advice. We have added a "Data Information"

section to each legend, providing details on the statistics used, information regarding replicates, and scales. Following the format provided in the link you shared, we have incorporated this information.

- We now use CRediT to specify the contributions of each author in the journal submission system. CRediT replaces the author contribution section. Please use the free text box to provide more detailed descriptions and do not provide your final manuscript text file with an author contributions section. See also our guide to authors:

<https://www.embopress.org/page/journal/14693178/authorguide#authorshipguidelines>

Re: We would like to express our gratitude for your kind assistance. With your guidance, we have promptly updated our submission to align with the CRediT system for specifying the contributions of each author. Additionally, we have removed the author contributions section from the manuscript text file as per your instructions.

- We updated our journal's competing interests policy in January 2022 and request authors to consider both actual and perceived competing interests. Please review the policy <https://www.embopress.org/competing-interests> and update your competing interests if necessary. Please name this section 'Disclosure and Competing Interests Statement' and put it after the Acknowledgements section.

Re: We would like to express our appreciation for your efforts in updating the competing interests policy. We have carefully reviewed and considered both actual and perceived competing interests in accordance with your guidance. We have included a section titled "Disclosure and Competing Interests Statement" following the Acknowledgements section, ensuring transparency and compliance.

- Please remove the sections "Corresponding author", the description of the synopsis image ("Synopsis: Graphical abstract of the main findings in this study" - BioRender can be acknowledged in the acknowledgements section) and the "Ethics declarations" from the manuscript text file. Or, if there are ethic declarations, please add these.

Re: We thank your guidance in streamlining the manuscript text file. With these, we have removed the "Corresponding author" section, the description of the synopsis image, and the "Ethics declarations" as per your instructions.

- Appendix Table S1 is a datasets. Please upload this as single excel file named Dataset EV1. Please add a title and a legend on the first TAB this file and please change the callouts of this item in the manuscript text file (Dataset EV1).

Re: We are thankful for your assistance in addressing this matter. We have promptly uploaded Appendix Table S1 as a single Excel file named Dataset EV1. We

sincerely appreciate your guidance in providing a title and a comprehensive legend on the first tab of the file. Additionally, we have meticulously updated the callouts in the manuscript text file to accurately reference Dataset EV1.

- Please make sure that all the funding information is also entered into the online submission system and that it is complete and similar to the one in the acknowledgement section of the manuscript text file. Presently, the Innovative Research Team of High-Level Local Universities in Shanghai is only mentioned in the acknowledgements.

Re: We express our gratitude for your attention to detail. We have promptly ensured that all the funding information is accurately entered into the online submission system. We are sincerely thankful for your guidance in reviewing the information to guarantee its completeness and consistency with the acknowledgement section of the manuscript text file. Specifically, we have ensured that the Innovative Research Team of High-Level Local Universities in Shanghai are included in online submission system.

- The Data Availability section should only contain information on large datasets that have been deposited to external repositories and all access information. Please remove any additional information from this section and make sure the datasets are public latest upon publication of the manuscript.

Re: We appreciate your guidance in revising the Data Availability section. We have removed any additional information, ensuring that the section exclusively contains details regarding large datasets deposited in external repositories. We remain committed to making these datasets publicly available upon the publication of the manuscript.

- Thanks for providing a schematic summary figure (synopsis image). Please provide this with bigger fonts and less blurry (but with the exact width of 550 pixels and a height of not more than 400 pixels).

Re: We are greatly thankful for your feedback on the synopsis image. Your suggestions have been immensely valuable to us. We have made the necessary adjustments, ensuring that the font size is increased, and the image's clarity is improved. Moreover, we have adhered to the requested dimensions of 550 pixels width and a height of not more than 400 pixels.

reviewers' comments

Referee #1:

The authors are very responsive and address many concerns. The manuscript is significantly improved. However, there is a concern: The images of HE staining in Figure 5C and 7C are very hard to appreciate.

Re: We are extremely grateful for acknowledging our responsiveness and the significant improvements made to the manuscript. We apologize for the inconvenience caused by the difficulty in appreciating the images of HE staining in Figures 5C and 7C. We have taken this feedback seriously, and we made effort to enhance the visibility and clarity of these images, The inflammatory cells are depicted with greater clarity in the final version of the manuscript. We sincerely thank Referee #1 for bringing this concern to our attention.

Referee #2:

the authors respond adequately to my requests.

Re: We would like to express our heartfelt thanks for recognizing our efforts in adequately addressing their requests and concerns. This feedback has been instrumental in shaping the manuscript, and we are grateful for Referee #2 for their valuable input and guidance.

Referee #3:

The authors have sufficiently addressed my concerns in this revision. One minor revision to further improve the quality of the manuscript is probably show flow cytometry analysis of the markers in Figure 4A as well.

Re: We express our utmost gratitude for the acknowledgement of our efforts in addressing the concerns raised during the revision process. We sincerely appreciate the valuable suggestion provided by Referee #3 to include flow cytometry analysis of the markers in Figure 4A, which has significantly enhanced the quality of the manuscript. Accordingly, we promptly incorporated this analysis into the final version of the manuscript (**revised Fig. 5C, 7C**) and extend our sincere thanks to Referee #3 for their insightful recommendation.

In response to the suggestion, we have included representative histograms and median fluorescence intensity plots illustrating the protein expression of TGF- β /Smad signaling downstream genes in CD4⁺ T cells (**revised Fig. 5C, 7C**). These genes, namely Foxp3, Tgfb1, Girt, Cxcr4, and Ctla4, are associated with the TGF- β /Smad signaling pathway and relevant to regulatory T (Treg) cells (Schlenner *et al*, 2012; Zhang *et al*, 2006; McHugh *et al*, 2002; Tang *et al*, 2017). We activated the signaling pathway through TGF- β stimulation and treated the cells with either Dleu2-17aa or scrPEP for 24 hours.

Consistent with the results obtained from quantitative PCR (qPCR), the protein expression levels were found to be elevated under Dleu2-17aa treatment compared to scrPEP, mirroring the observed mRNA expression patterns.

We sincerely appreciate the acknowledgment of these additions and their contribution to the manuscript's overall improvement.

References

- McHugh RS, Whitters MJ, Piccirillo CA, Young DA, Shevach EM, Collins M & Byrne MC (2002) CD4+CD25+ Immunoregulatory T Cells: Gene Expression Analysis Reveals a Functional Role for the Glucocorticoid-Induced TNF Receptor. *Immunity* 16: 311–323
- Schlenner SM, Weigmann B, Ruan Q, Chen Y & von Boehmer H (2012) Smad3 binding to the foxp3 enhancer is dispensable for the development of regulatory T cells with the exception of the gut. *J Exp Med* 209: 1529–1535
- Tang PM-K, Zhou S, Meng X-M, Wang Q-M, Li C-J, Lian G-Y, Huang X-R, Tang Y-J, Guan X-Y, Yan BP-Y, *et al* (2017) Smad3 promotes cancer progression by inhibiting E4BP4-mediated NK cell development. *Nat Commun* 8: 14677
- Zhang M, Fraser D & Phillips A (2006) ERK, p38, and Smad Signaling Pathways Differentially Regulate Transforming Growth Factor- β 1 Autoinduction in Proximal Tubular Epithelial Cells. *The American Journal of Pathology* 169: 1282–1293

Prof. Honglin Wang
Precision Research Center for Refractory Diseases
Shanghai General Hospital
650 Xinsongjiang Road
Shanghai 200025
China

Dear Prof. Wang,

I am very pleased to accept your manuscript for publication in the next available issue of EMBO reports. Thank you for your contribution to our journal.

Yours sincerely,
